# Deep Multi-Marginal Momentum Schrödinger Bridge

**Tianrong Chen, Guan-horng Liu, Molei Tao, Evangelos A. Theodorou**
Georgia Institute of Technology, USA
`{tianrong.chen,ghliu, mtao, evangelos.theodorou}@gatech.edu`

## Abstract

It is a crucial challenge to reconstruct population dynamics using unlabeled samples from distributions at coarse time intervals. Recent approaches such as flow-based models or Schrödinger Bridge (SB) models have demonstrated appealing performance, yet the inferred sample trajectories either fail to account for the underlying stochasticity or are unnecessarily rigid. In this article, We extend the approach in [1] to operate in continuous space and propose Deep Momentum Multi-Marginal Schrödinger Bridge (DMSB), a novel computational framework that learns the smooth measure-valued spline for stochastic systems that satisfy position marginal constraints across time. By tailoring the celebrated Bregman Iteration and extending the Iteration Proportional Fitting to phase space, we manage to handle high-dimensional multi-marginal trajectory inference tasks efficiently. Our algorithm outperforms baselines significantly, as evidenced by experiments for synthetic datasets and a real-world single-cell RNA sequence dataset. Additionally, the proposed approach can reasonably reconstruct the evolution of velocity distribution, from position snapshots only, when there is a ground truth velocity that is nevertheless inaccessible.

## 1   Introduction

We consider the multi-marginal trajectory inference problem, which pertains to elucidating the dynamics and reactions of indiscernible individuals, given static snapshots of them taken at sporadic time points. Due to the inability of tracking each individual, one considers the evolution of the statistical distribution of the population instead. This problem received considerable attention, and associated applications appear in various scientific areas such as estimating cell dynamics [2, 3], predicting meteorological evolution [4], and medical healthcare statistics tracking [5]. [6, 7] constructed an energy landscape that best aligned with empirical observations using neural network. [8, 9] learn regularized Neural ODE [10] to encode such potential landscape. Notably, in the aforementioned work, the trajectory of samples is represented in a deterministic way. In contrast, [11, 12] employ Schrödinger Bridge (SB) to determine the most likely evolution of samples between marginal distributions when individual sample trajectories are also affected by environmental stochasticity. Yet, these approaches scale poorly w.r.t. the state dimension due to specialized neural network architectures and computational frameworks.

SB can be viewed as a solution to the entropy-regularized optimal transport problem. SB seeks a nonlinear SDE that yields a straight path measure between two *arbitrary* distributions. The straightness is implied by achieving optimality of minimizing transportation costs (i.e. 2-Wasserstein distance ($W_2$)). We note SB is often related to Score-based Generated Model (SGM), both of which can be used for generative modeling by constructing certain Stochastic Differential Equation (SDE) that links data distribution and a tractable prior distribution (i.e. 2 marginals). SGM accomplishes the generative task by first diffusing data to prior through a pre-specified linear SDE, during which a neural network is also learned to approximate the score function. Then this score approximator is used to reverse this diffusion process, and consequently establish the generation. Critically-damped

Table 1: Comparison between different models in terms of optimality and boundary distributions $p_0$ and $p_1$. Our DMSB extends standard SB, which generalizes SGM beyond Gaussian priors, to phase space, similar to CLD. However, unlike CLD, DMSB jointly *learns* the phase space distributions, i.e., $p_\theta(x, v) = p_{\mathcal{A}}(x)q_\theta(v|x)$ and $p_\phi(x, v) = p_{\mathcal{B}}(x)q_\phi(v|x)$. In other words, DMSB infers the underlying phase state dynamics given only state distributions.

| Models | Optimality | $p_0(\cdot)$ | $p_1(\cdot)$ |
|--------|-----------|--------------|--------------|
| SGM [20] | ✗ | $p_{\mathcal{A}}(x)$ | $\mathcal{N}(\mathbf{0}, \mathbf{\Sigma})$ |
| CLD [13] | ✗ | $p_{\mathcal{A}}(x) \otimes \mathcal{N}(\mathbf{0}, \mathbf{\Sigma})$ | $\mathcal{N}(\mathbf{0}, \mathbf{\Sigma}) \otimes \mathcal{N}(\mathbf{0}, \mathbf{\Sigma})$ |
| SB [14] | $\downarrow W_2 \to$ kinks | $p_{\mathcal{A}}(x)$ | $p_{\mathcal{B}}(x)$ |
| DMSB (ours) | $\downarrow W_2 \to$ smooth | $p_{\mathcal{A}}(x)q_\theta(v|x)$ | $p_{\mathcal{B}}(x)q_\phi(v|x)$ |

Langevin Diffusion (CLD) [13] extends the SGM SDE to the phase space by introducing an auxiliary velocity variable with a tractable Gaussian distribution at both the initial and terminal time. The resulting trajectory in the position space becomes smoother, as stochasticity is only injected into the velocity space, and the empirical performance and sample efficiency are enhanced due to the structure of the critical damped SDE. The connection between SGM and SB has been elaborated in [14, 15] and scalable mean matching Iterative Proportional Fitting algorithm (IPF) is proposed to estimate SB efficiently in high dimensional cases. Applications of SB, such as image-to-image transformation [16, 17], RNA trajectory inference [11], solving Mean Field Game[18], Riemannian interpolation [19], demonstrate the effectiveness of SB in various domains.

In this work, we start with SB in phase space (termed momentum SB, mSB in short), and then further investigate mSB with multiple empirical marginal constraints present in the position space, which was formulated as multi-marginal mSB (mmmSB) in [1]. This circumvents the need for expensive space discretization which does not scale well to high dimensions. We also address the challenge of intricate geometric averaging in continuous space setup by strategically partitioning and reorganizing the constraint sets. Furthermore, we enhance the algorithm's computational efficiency by incorporating the method of half-bridge IPF. The optimality of transportation cost in SB leads to straight trajectories, and if one solves N 2-marginal SB problems and connect the resulting trajectories to match N+1 marginals, the connected trajectories will have kinks at all connection points. On the contrary, in mmmSB, the optimality of transportation cost leads to a smooth measure-spline over the state space that also interpolates the empirical marginals. Therefore, this approach is highly suitable for problems originated from physical systems and/or those that should have smooth trajectories, such as trajectory inference in single-cell RNA sequencing. Our research will emphasize on solving mmmSB efficiently in high-dimensions (thus the approach will differ from that in the seminal work [1]; see Sec.4). The differences between our algorithm and prior work are demonstrated in Table.1, and the main contributions of our work are fourfold:

- We extend the mean matching IPF to phase space allowing for scalable mSB computing.

- We introduce and tailor the Bregman Iteration [21] for mmmSB which makes it compatible with the phase space mean matching objective, thus the efficient computation is activated for high dimensional mmmSB.

- We show how to overcome the challenge of sampling the velocity variable when it is not available in training data, which enhances the applicability of our model.

- We show the performance of proposed algorithm DMSB on toy datasets which contains intricate bifurcations and merge. On realistic high-dimension (100-D) single-cell RNA-seq (scRNA-seq) datasets, DMSB outperforms baselines by a significant margin in terms of the quality of the generated trajectory both visually and quantitatively. We show that DMSB is able to capture reasonable velocity distribution compared with ground truth while other baselines fail.

## 2 Preliminary

### 2.1 Dynamical Schrödinger Bridge problem

Dynamical Schrödinger Bridge problem has been extensively studied in the past few decades. The objective of the SB problem is to solve the following optimization problem:

$$\min_{\pi \in \Pi(\rho_0, \rho_T)} D_{KL}(\pi||\xi), \tag{1}$$

where $\pi \in \Pi(\rho_0, \rho_T)$ belongs to a set of path measures with its marginal densities at $t = 0$ and $T$ being $\rho_0$ and $\rho_T$. $\xi$ is the reference path measure (i.e., [14] sets $\xi$ as Wiener process from $\rho_0$). The optimality of the problem (1) is characterized by a set of PDEs (3).

**Theorem 2.1** ([22]). *The optimal path measure $\pi$ in the problem (1) is represented by forward and backward stochastic processes*

$$d\mathbf{x}_t = [2 \nabla_\mathbf{x} \log \Psi_t]dt + \sqrt{2} d\mathbf{w}_t, \quad \mathbf{x}_0 \sim \rho_0, \tag{2a}$$

$$d\mathbf{x}_t = [-2 \nabla_\mathbf{x} \log \widehat{\Psi}_t]dt + \sqrt{2} d\widehat{\mathbf{w}}_t, \mathbf{x}_T \sim \rho_T. \tag{2b}$$

*in which $\Psi, \widehat{\Psi} \in C^{1,2}$ are the solutions to the following coupled PDEs,*

$$\frac{\partial \Psi_t}{\partial t} = -\Delta \Psi_t, \quad \frac{\partial \widehat{\Psi}_t}{\partial t} = \Delta \widehat{\Psi}_t \tag{3}$$

$$s.t. \ \Psi(0, \cdot)\widehat{\Psi}(0, \cdot) = \rho_0(\cdot), \ \Psi(T, \cdot)\widehat{\Psi}(T, \cdot) = \rho_T(\cdot),$$

The stochastic processes of SB in (2a) and (2b) are equivalent in the sense of $\forall t \in [0, T], p_t^{(2a)} \equiv p_t^{(2b)} \equiv p_t^{SB}$. Here $p_t^{SB}$ stands for the marginal distribution of SB at time $t$, which also represents the marginal density of stochastic process induced by either of Eq.2. The potentials $\Psi_t$ and $\widehat{\Psi}_t$ explicitly represent the solution of Fokker-Plank Equation (FPE) and Hamilton–Jacobi–Bellman equation (HJB) after exponential transform [14] where FPE describes the evolution of samples density and HJB represents for the optimality of Eq.1. Furthermore, the marginal density also obeys a factorization of $p_t^{SB} = \Psi_t\widehat{\Psi}_t$. Such rich structures of SB will later on be used to construct the log-likelihood objective (Thm.B.1) and Langevin sampler for velocity (§4.4).

To solve SB, prior work have primarily used the half-bridge optimization technique, also known as Iterative Proportional Fitting (IPF), in which one iteratively solves the optimization problem with one of the two boundary conditions [14, 15, 23],

$$\pi^{(d+1)} := \arg\min_{\pi \in \Pi(\cdot, \rho_1)} D_{KL}(\pi||\pi^{(d)}) \quad \rightleftarrows \quad \pi^{(d+2)} := \arg\min_{\pi \in \Pi(\rho_0, \cdot)} D_{KL}(\pi||\pi^{(d+1)}) \tag{4}$$

with initial path measure $\pi^{(0)} := \xi$. By repeatedly iterating over aforementioned optimizations until the algorithm converges, the SB solution will be attained as $\pi^{SB} \equiv \lim_{d\to\infty} \pi^{(d)}$ [24]. In addition, [25] shows that the drift term in SB problem can also be interpreted as the solution Stochastic Optimal Control (SOC) problem by having optimal control policy $\mathbf{z}^* = 2 \nabla_\mathbf{x} \log \Psi(t, \mathbf{x}_t)$:

$$\mathbf{z}^*(\mathbf{x}) \in \arg\min_{\mathbf{z} \in \mathcal{Z}} \mathbb{E}\left[\int_0^T \frac{1}{2}\|\mathbf{z}_t\|^2 dt\right] \quad s.t \quad \begin{cases} d\mathbf{x}_t = \mathbf{z}_t dt + \sqrt{2}d\mathbf{w}_t \\ \mathbf{x}_0 \sim \rho_0, \quad \mathbf{x}_1 \sim \rho_T. \end{cases}$$

This formulation will be used later on for constructing phase space likelihood objective function in §3. Regarding solving the half-bridge problem, abundant results exist in the literature for the vanilla SB described above [14, 15, 23], but we will be solving a different SB problem; see Prop.4.1 for formulation and §.4 for a solution.

### 2.2 Bregman Iterations for Multiple Constraints

Bregman iteration [21] can be viewed as a multiple marginal generalization of IPF, and it is widely used to solve entropy regularized optimal transport problem [1] with multiple constraints. The algorithm can efficiently solve problems in the form of,

$$\inf_{\pi \in \mathcal{K}} KL(\pi|\xi),$$

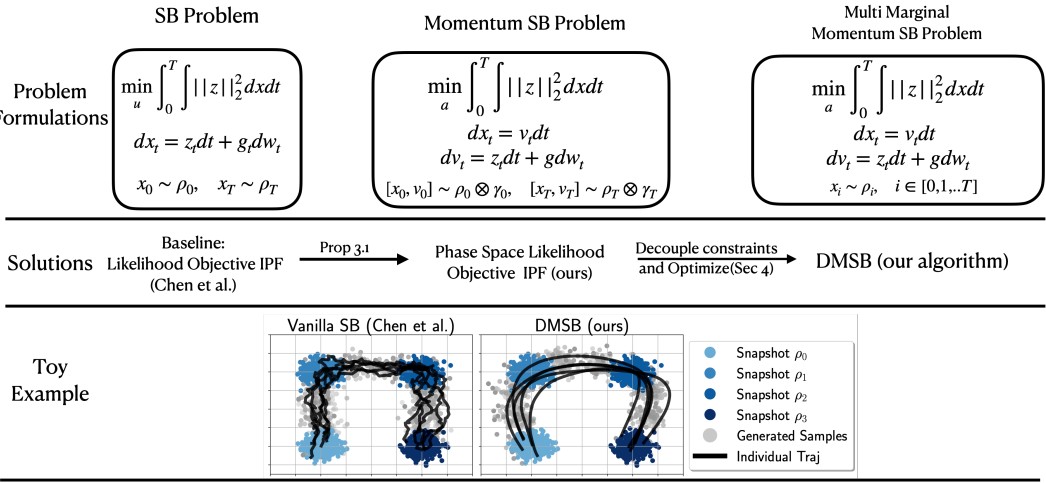

Figure 1: A summary of various SB problems and corresponding algorithms. The toy example in the 3rd row illustrates that vanilla SB determines 'straight' paths (modulo fluctuations due to noise) between pairwise empirical marginals, while our multi-marginal momentum SB approach establishes a smooth measure-spline between marginals in the position space (albeit still stochastic, the path is smooth between any pair of adjacent 2 marginals, because noise is added to velocity, and the path is also smooth across different pairs of adjacent 2 marginals per design.

where $\mathcal{K}$ is the intersection of multiple closed convex constraint sets $\mathcal{K}_l$: $\mathcal{K} = \cap_{l=1}^{L} \mathcal{K}_l$. Bregman Projection (BP) is defined as optimization w.r.t one of the constraint $\mathcal{K}_l$,

$$P_{\mathcal{K}_l}^{KL}(\xi) := \arg\min_{\pi \in \mathcal{K}_l} KL(\pi|\xi),$$

and $d$-th Bregman Iteration (BI) is recursively computing BP over all the constraints in $\mathcal{K}$:

$$\forall 0 < n \leq L, \quad \pi^{(d,n)} := P_{\mathcal{K}_l^n}^{KL}(\pi^{(d,n-1)}),$$

The initial condition for $(d+1)$-th BI is $\pi^{(d+1,0)} = \pi^{(d,L)}$. Under certain conditions (see e.g., [24]), one has that $\pi^{(d,L)}$ converges to the unique solution:

$$\pi^{(d,L)} \to P_{\mathcal{K}}^{KL}(\xi) \quad \text{as} \quad d \to +\infty$$

**Remark 2.2.** One BI traverses all constraints via multiple BPs, and each BP solves an optimization problem with one constraint. One can notice that the BI will become the aforementioned IPF procedure solving SB problem (1) by defining $L = 2, \mathcal{K}_1 = \Pi(\rho_0, \cdot), \mathcal{K}_2 = \Pi(\cdot, \rho_1)$.

Table 2: Mathematical notation.

| Notation | Definition | Notation | Definition |
|---|---|---|---|
| $\mathbf{x}$ | position variable | $\rho$ | position distribution $\rho(\mathbf{x})$ |
| $\mathbf{v}$ | velocity variable | $\gamma$ | velocity Distribution $\gamma(\mathbf{v})$ |
| $\mathbf{m}$ | concatenation of $[\mathbf{x}, \mathbf{v}]^{\mathsf{T}}$ | $\mu$ | distribution of $\mu(\mathbf{x}, \mathbf{v})$ |

## 3 Momentum Schrödinger Bridge

We first describe how to conduct half-bridge IPF training in the phase space, which can be used to solve momentum SB (mSB) problem with two marginals constraints. This scalable phase space half-bridge technique will then be applied to multi-marginal cases (Sec.4). Fig.1 demonstrates how we develop an algorithm based on [14]. Notations used in following sections are listed in Table.2. mSB extends SB problem to phase space, which consists of both position and velocity.

We will first consider boundary distributions that depend on both $\mathbf{x}$ and $\mathbf{v}$, although eventually we will use this as a module to find transport maps between two distributions that only depend on position $\mathbf{x}$, as velocity $\mathbf{v}$ is an auxiliary variable artificially introduced for obtaining smooth transport. Conceptually, as an entropy regularized optimal transport problem, SB tries to obtain the straightest path between empirical marginals of positions $\mathbf{x}$ with additive noise, but mSB aims at finding the smooth interpolation between empirical marginals of $\mathbf{x}$ [26] conditioned on boundary velocity distributions (see Fig.1). Such smooth measure-valued splines in the position space are obtained by the optimization problem in the phase space [1]:

$$\min_{\pi \in \Pi(\mu_0, \mu_T)} KL(\pi|\xi) \quad s.t \quad \pi = \mathrm{Law}(\mathbf{x}, \mathbf{v}): \underbrace{\begin{pmatrix} \mathrm{d}\mathbf{x}_t \\ \mathrm{d}\mathbf{v}_t \end{pmatrix}}_{\mathrm{d}\mathbf{m}_t} = \underbrace{\begin{pmatrix} \mathbf{v}_t \\ \mathbf{0} \end{pmatrix}}_{\boldsymbol{f}(\mathbf{v}, t)} \mathrm{d}t + \underbrace{\begin{pmatrix} \mathbf{0} & \mathbf{0} \\ \mathbf{0} & g_t \end{pmatrix}}_{\mathbf{g}(t)} \underbrace{\begin{pmatrix} \mathbf{0} \\ \mathbf{z}_t \end{pmatrix}}_{\mathbf{Z}(t)} \mathrm{d}t + \underbrace{\begin{pmatrix} \mathbf{0} & \mathbf{0} \\ \mathbf{0} & g_t \end{pmatrix}}_{\mathbf{g}(t)} \mathrm{d}\mathbf{w}_t,$$

Similar to Theorem 2.1, one can derive a set of PDEs using the potential functions $\Psi(t, \mathbf{x}, \mathbf{v})$ and $\widehat{\Psi}(t, \mathbf{x}, \mathbf{v})$, and subsequently apply IPF procedure to solve the problem. The formulation of the phase space PDE can be found in Appendix.B.2. Such PDE representation of mSB results in a straightforward yet innovative log-likelihood training that enables efficient optimization of the IPF.

**Proposition 3.1** (likelihood bound). *The half-bridge IPF in phase space*

$$\pi^{(d+1)} := \operatorname*{arg\,min}_{\pi \in \Pi(\mu_0, \cdot)} D_{KL}(\pi || \pi^{(d)}) \quad \rightleftarrows \quad \pi^{(d+2)} := \operatorname*{arg\,min}_{\pi \in \Pi(\cdot, \mu_T)} D_{KL}(\pi || \pi^{(d+1)})$$

*represents the bound of the likelihood and gives approximate likelihood training:*

$$\mathbf{Z}_t := \operatorname*{arg\,min}_{\widehat{\mathbf{Z}}_t} -\log p(\mathbf{m}_0, 0) \quad \rightleftarrows \quad \widehat{\mathbf{Z}}_t := \operatorname*{arg\,min}_{\widehat{\mathbf{z}}_t} -\log p(\mathbf{m}_T, T).$$

$$where \quad \log p(\mathbf{m}_0, 0) \propto \int_0^T \mathbb{E}_{\widehat{\mathbf{m}}_t} \left[ \frac{1}{2} \|\widehat{\mathbf{z}}_t + \mathbf{z}_t - g\nabla_\mathbf{v} \log \hat{p}_t\|^2 \right] \mathrm{d}t.$$

*and $\widehat{\mathbf{m}}_t$ samples from:* $\quad \mathrm{d}\widehat{\mathbf{m}}_t = \left[ \boldsymbol{f} - \mathbf{g}\widehat{\mathbf{Z}}_t \right] \mathrm{d}t + \mathbf{g}(t)\mathrm{d}\mathbf{w}_t, \quad \widehat{\mathbf{m}}_T \sim \mu_T$ (5)

$\widehat{\mathbf{Z}}_t \triangleq \begin{pmatrix} \mathbf{0} \\ \widehat{\mathbf{z}}_t \end{pmatrix}$ *and $\widehat{p}_t$ is the density of path measure induced by eq.5 at time $t$. A similar result for* $\log(\mathbf{m}_T, T)$ *can be obtained in a similar derivation.*

*Proof.* See Appendix B.1. $\qquad\square$

**Remark 3.2.** After optimizing $\widehat{\mathbf{Z}}_t$, the reference path measure becomes eq.5, which implies $\pi \in \Pi(\cdot, \mu_T)$, i.e., the constraint in half-bridge IPF is satisfied. A path measure $\pi$ is induced by either $\mathbf{Z}_t$ or $\widehat{\mathbf{Z}}_t$. As being mentioned in Remark.2.2. One half-bridge IPF is basically one BP and one IPF is one BI. Prop.3.1 provides a convenient way to perform one BP in the form of $\pi := \arg\min_{\pi \in \mathcal{K}_l} D_{KL}(\pi || \bar{\pi})$ by maximizing log-likelihood given constraint $\mathcal{K}$ and reference path measure $\bar{\pi}$.

Prop.3.1 provides an alternative way to conduct the BI which will be heavily used in mmmSB §3, and it is computationally efficient after parameterizing and discretization (§4.4).

# 4 Deep Momentum Multi-Marginal Schrödinger Bridge

We first state the problem formulation of momentum multi-marginal Schrödinger Bridge (mmmSB). Different from previous two marginals case, we consider the scenario where $N + 1$ probability measures $\mu_{t_i}$ are lying at time $t_i$. In addition, velocity distributions are not necessarily known.

**Proposition 4.1** ([1]). *The dynamical mmmSB with multiple marginal constraints reads:*

$$\min_\pi \mathcal{J}(\pi) := \sum_{i=0}^{N-1} KL\left(\pi_{t_i:t_{i+1}} | \xi_{t_i:t_{i+1}}\right), \quad s.t \quad \pi \in \mathcal{K} := \cap_{i=0}^N \mathcal{K}_{t_i}$$ (6)

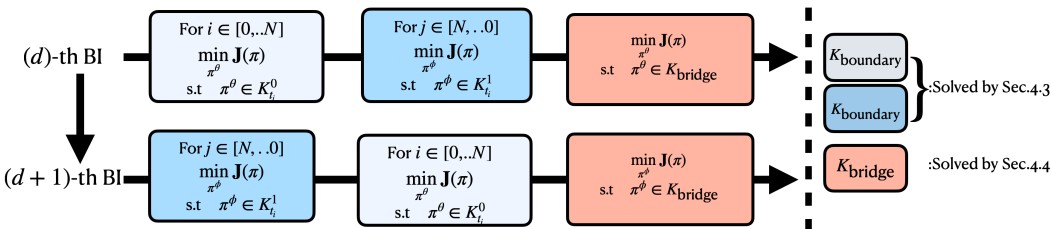

Figure 2: The procedure details the Bregman Iteration (BI) employed in DMSB. The gray and blue blocks represent the BP step performed under $\mathcal{K}_{\text{boundary}}$ constraint for forward and backward policies, respectively. The red block signifies the BP step executed under the $\mathcal{K}_{\text{bridge}}$ constraint. Algorithms for training and sampling can be found in Appendix.D.

$$\textit{where:} \quad \mathcal{K}_{t_0} = \left\{ \int \pi_{t_0:t_1} \mathbf{dm}_{t_1} = \mu_{t_0}, \int \mu_{t_0} \mathbf{dv}_{t_0} = \rho_{t_0} \right\}$$

$$\mathcal{K}_{t_N} = \left\{ \int \pi_{t_{N-1}:t_N} \mathbf{dm}_{t_{N-1}} = \mu_{t_N}, \int \mu_{t_N} \mathbf{dv}_{t_N} = \rho_{t_N} \right\}$$

$$\mathcal{K}_{t_i} = \left\{ \int \pi_{t_i:t_{i+1}} \mathbf{dm}_{t_{i+1}} = \mu_{t_i}, \int \pi_{t_{i-1}:t_i} \mathbf{dm}_{t_{i-1}} = \mu_{t_i}, \int \mu_{t_i} \mathbf{dv}_{t_i} = \rho_{t_i} \right\}, \quad (7)$$

*and $\mathcal{K}$ is the intersection of close convex set of $\mathcal{K}_{t_i}$.*

The problem described in Prop.4.1 can be solved by classical BI algorithm integrated with Sinkhorn method [1]. However, due to the curse of dimensionality and unfavorable geometric explicit solution, the BP cannot be applied in high-dimensional and continuous state space directly. To tackles these difficulties, we parameterize the forward and backward policies $\mathbf{z}_t$ and $\widehat{\mathbf{z}}_t$ by a pair of neural networks. We further decouple and resemble the constraints by which it enables the scalable likelihood IPF and avoids the geometric averaging issue under mmmSB context.

## 4.1 Decoupling and Reassembling Constraints

We decompose the constraint set (7) by

$$\mathcal{K}_{t_i} = \cap_{r=0}^2 \mathcal{K}_{t_i}^r, \quad \text{where} \quad \begin{matrix} \mathcal{K}_{t_i}^0 = \left\{ \int \pi_{t_i:t_{i+1}} \mathbf{dm}_{t_{i+1}} = \hat{\mu}_{t_i}, \int \hat{\mu}_{t_i} \mathbf{dv}_{t_i} = \rho_{t_i} \right\} \\ \mathcal{K}_{t_i}^1 = \left\{ \int \pi_{t_{i-1}:t_i} \mathbf{dm}_{t_{i-1}} = \mu_{t_i}, \int \mu_{t_i} \mathbf{dv}_{t_i} = \rho_{t_i} \right\} \\ \mathcal{K}_{t_i}^2 = \left\{ \int \pi_{t_i:t_{i+1}} \mathbf{dm}_{t_{i+1}} = \int \pi_{t_{i-1}:t_i} \mathbf{dm}_{t_{i-1}} \right\}. \end{matrix} \quad (8)$$

One can notice that the $\mathcal{K}_{t_i}^0$ and $\mathcal{K}_{t_i}^1$ share similar structure as simpler boundary marginal conditions $\mathcal{K}_{t_0}$ and $\mathcal{K}_{t_N}$, hence we can get rid of the notorious geometric averaging (see §4 in [1]). Notably, this type of constraint provides an opportunity to utilize Proposition 3.1 for optimization, but the joint distribution of $\mathbf{x}$ and $\mathbf{v}$ is still absent. We classify the constraints into two categories:

$$\mathcal{K}_{\text{boundary}} = \left\{ \cap_{i=1}^{N-1} \mathcal{K}_{t_i}^r \cap \mathcal{K}_{t_0} \cap \mathcal{K}_{t_N} | \forall r \in \{0,1\} \right\}, \quad \mathcal{K}_{\text{bridge}} = \left\{ \cap_{i=1}^{N-1} \mathcal{K}_{t_i}^2 \right\}.$$

By following BI (§2.2), we execute optimization w.r.t. (6) while projecting the solution to subset of $\mathcal{K}_{\text{boundary}}$ or $\mathcal{K}_{\text{bridge}}$ iteratively. The sketch can be found in Fig.2. The next sections will provide more details on obtaining the joint distribution $\mu$ and optimizing within each constraint set.

Hereafter, we only demonstrate the optimization for forward policy $\mathbf{z}_t$ given reference path measure $\bar{\pi}$ driven by fixed backward policy $\widehat{\mathbf{z}}_t$. The procedure can be applied for the $\widehat{\mathbf{z}}_t$ and vice versa.

## 4.2 Optimization in set $\mathcal{K}_{\text{boundary}}$

We first show how to optimize forward policy $\mathbf{z}_t$ w.r.t. objective function (6) given the reference path measure $\bar{\pi}$ driven by fixed backward policy $\widehat{\mathbf{z}}_t$ under one subset of $\mathcal{K}_{\text{boundary}}$.

**Proposition 4.2** (Optimality w.r.t. $\mathcal{K}_{\text{boundary}}$). *Given the reference path measure $\bar{\pi}$ driven by the backward policy $\widehat{\mathbf{z}}_t$ from boundary $\mu_{t_{i+1}}$ in the reverse time direction, the optimal path measure in the forward time direction of the following problem*

$$\min_{\pi} \mathcal{J}(\pi) := \sum_{i=0}^{N-1} KL\left(\pi_{t_i:t_{i+1}} | \bar{\pi}_{t_i:t_{i+1}}\right), \quad s.t \quad \pi \in \left\{ \int \pi_{t_i:t_{i+1}} \mathbf{dm}_{t_{i+1}} = \mu_{t_i}, \int \mu_{t_i} \mathbf{dv}_{t_i} = \rho_{t_i} \right\}$$

$$is: \quad \pi^*_{t_i:t_{i+1}} = \frac{\rho_{t_i} \bar{\pi}_{t_i:t_{i+1}}}{\int \bar{\pi}_{t_i:t_{i+1}} \mathbf{dm}_{t_{i+1}} \mathbf{dv}_{t_i}}.$$

When $\pi_{t_i:t_{i+1}} \equiv \pi^*_{t_i:t_{i+1}}$, the following equations need to hold $\forall t \in [t_i, t_{i+1}]$:

$$\|\mathbf{z}_t + \widehat{\mathbf{z}}_t - g\nabla_{\mathbf{v}} \log \hat{p}_t\|_2^2 = 0, \tag{9a}$$

$$p_{t_i}(\mathbf{v}_{t_i}|\mathbf{x}_{t_i}) \equiv \hat{q}_{t_i}(\mathbf{v}_{t_i}|\mathbf{x}_{t_i}), \tag{9b}$$

where $\hat{p}_t$ and $\hat{q}_t$ denote the marginal density and conditional velocity distribution of the reference path measure at time $t$, respectively.

*Proof.* See appendix.B.5 $\qquad\qquad\qquad\qquad\qquad\qquad\qquad\qquad\qquad\qquad\qquad$ $\square$

**Remark 4.3.** When the ground truth distributions of velocity $\gamma_{t_i}$ are available, one can simply sample from $\gamma_{t_i}$ since the joint distribution $\mu_t$ is available in this case. In order to matching the reference path measure in KL divergence sense, one needs to match both the intermediate path measure eq.9a and the boundary condition eq.9b. In the traditional two-boundary SB case, matching the boundary condition is often disregarded due to either having a predefined data distribution or a tractable prior. However, in our specific case, as the velocity is not predefined, it becomes imperative to address this issue and optimize it through the application of Langevin dynamics.

### 4.3 Optimization in set $\mathcal{K}_{\text{bridge}}$

The formulation of optimization under $\mathcal{K}_{\text{bridge}}$ is similar to the previous section but differs by the boundary condition (eq.10b):

**Proposition 4.4** (Optimality w.r.t. $\mathcal{K}_{\text{bridge}}$). *Given the reference path measure $\bar{\pi}$ driven by the backward policy $\widehat{\mathbf{z}}_t$ from boundary $\mu_{t_N}$ in the reverse time direction, the optimal path measure in the forward time direction of the following problem*

$$\min_{\pi} \mathcal{J}(\pi) := \sum_{i=0}^{N-1} KL\left(\pi_{t_i:t_{i+1}}|\bar{\pi}_{t_i:t_{i+1}}\right), \quad s.t \quad \pi \in \mathcal{K}_{bridge} = \left\{\cap_{i=1}^{N-1} \mathcal{K}_{t_i}^2\right\}$$

$$is: \quad \pi^*_{t_0:t_N} = \frac{q_{t_0} \bar{\pi}_{t_0:t_N}}{\int \bar{\pi}_{t_0:t_N} \mathbf{dm}_{t_N} \mathbf{dv}_{t_0}}.$$

*when $\pi_{t_0:t_N} \equiv \pi^*_{t_0:t_N}$, the following equations need to hold $\forall t \in [t_0, t_N]$:*

$$\|\mathbf{z}_t + \widehat{\mathbf{z}}_t - g\nabla_{\mathbf{v}} \log \hat{p}_t\|_2^2 = 0 \tag{10a}$$

$$p_{t_0}(\mathbf{v}_{t_0}, \mathbf{x}_{t_0}) \equiv \hat{q}_{t_0}(\mathbf{v}_{t_0}, \mathbf{x}_{t_0}) \tag{10b}$$

*Proof.* See appendix.B.6 $\qquad\qquad\qquad\qquad\qquad\qquad\qquad\qquad\qquad\qquad\qquad$ $\square$

Conceptually, the above optimization objective with $\mathcal{K}_{\text{bridge}}$ constraint aims at finding a *continuous* path measure close to reference path measure $\bar{\pi}$ while any intermediate marginals constraints will not be considered. The boundary condition of reference path measure in the next iteration $p_{t_0}(\mathbf{v}_{t_0}, \mathbf{x}_{t_0})$ is determined by eq.10b. Fortunately, the empirical samples from this distribution are available, though the analytic representation of the distribution $\hat{q}_{t_0}(\mathbf{v}_{t_0}, \mathbf{x}_{t_0})$ is unknown. Hence we can utilize these samples as empirical sources from boundary distribution $\hat{q}_{t_0}(\mathbf{v}_{t_0}, \mathbf{x}_{t_0})$ for the next BP. For further explanation and intuition, one can find it in Appendix.G

### 4.4 Parameterization and Training Objective Function

Inspired by the success of prior work [14], we parameterize path measure $\pi$ by forward policy $\mathbf{z}_t^\theta$ or backward policy $\widehat{\mathbf{z}}_t^\phi$ combined with one of constraints in $\mathcal{K}_{\text{boundary}}$ or $\mathcal{K}_{\text{bridge}}$ (see Fig.8 in Appendix for visualization). We adopt Euler–Maruyama discretization and denote the timestep as $\delta_t$. Notably, eq.9b and eq.10b can be implied by minimizing phase space NLL in Prop.3.1. This leads to the following objective function, termed as phase space mean matching objective, which will be used to train neural networks that represent $\mathbf{z}_t^\theta$ and $\widehat{\mathbf{z}}_t^\phi$ after time discretization:

$$\mathcal{L}_{MM} = \mathbb{E}\left[\|\delta_t \mathbf{z}_t^\theta(\mathbf{m}_{t+\delta_t}) + \delta_t \widehat{\mathbf{z}}_{t+\delta_t}^\phi(\mathbf{m}_{t+\delta_t}) - \left(\mathbf{m}_t + \delta_t \mathbf{z}_t^\theta - \mathbf{m}_{t+\delta_t}\right)\|^2\right].$$

The velocity boundary condition for the reference path measure in the succeeding BP is encoded in eq.9b or eq.10b, but the representation of conditional distribution eq.9b is not clear. We leverage the favorable property of SB to parameterize and sample from such distribution.

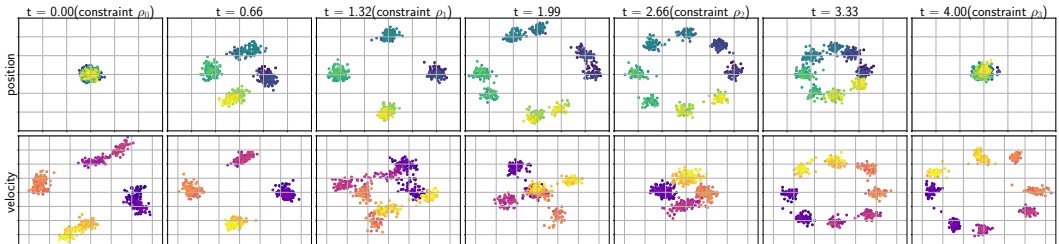

Figure 4: Validation of our DMSB model on complex GMM synthetic dataset. The velocity and position of the same sample correspond to the same shade level. *Upper*: Samples' evolution in the position space. *Bottom*: Learnt samples' evolution in the velocity space.

**Proposition 4.5** ([27, 28]). *If $\pi^\theta$ and $\bar{\pi}^\phi$ shares same path measure, then*

$$\tilde{p}_{t_i}^{\theta,\phi}(\mathbf{v}_{t_i}, \mathbf{x}_{t_i}) \equiv q_{t_i}^{\phi}(\mathbf{v}_{t_i}, \mathbf{x}_{t_i}) \propto q_{t_i}^{\phi}(\mathbf{v}_{t_i}|\mathbf{x}_{t_i}), \quad \textit{where:} \quad \nabla_{\mathbf{v}} \log \tilde{p}_t^{\theta,\phi} = \left(\mathbf{z}_t^\theta + \widehat{\mathbf{z}}_t^\phi\right)/g. \quad (11)$$

Prop.4.5 suggests that one can use $p_{t_i}(\mathbf{v}_{t_i}|\mathbf{x}_{t_i}) := \tilde{p}_{t_i}^{\theta,\phi}$ to imply condition (9b) and obtain samples from such distribution by simulating Langevin dynamics. Namely, we first sample position from ground truth $\mathbf{x}_{t_i} \sim \rho_{t_i}$, and then sample $\mathbf{v}_{t_i} \sim \tilde{p}_t^{\theta,\phi}$ using eq.11. One can further adopt the same regularization [29] to enforce the condition of Prop.4.5.

### 4.5 Training Scheme

Here we introduce the scheme to traverse BI (see Fig.2). In one BI, all constraints must be iterated once. For the sake of $\mathcal{L}_{\mathrm{MM}}$, the reference path measure should be induced by opposite direction. A single BI cannot be recursively repeated due to the conflict of reference path measure direction. For example (see Fig.2), at the end of $d$-th BI, $\bar{\pi}$ is yielded by forward policy while the first BP of $d$-th BI is also optimizing forward policy which violates $\mathcal{L}_{\mathrm{MM}}$. Instead, we reschedule the optimization order. Specifically, in $(d+1)$-th BI, we optimize backward policy at the first BP and the last BP.

## 5   Experiments

**Setups:** We test DMSB on 2D synthetic datasets and real-world scRNA-seq dataset [30]. We choose state of the art algorithms MIOFlow [9] and NLSB [11] as our baselines. We tune both models to the best of our hardware capacity. We choose Sliced-Wasserstein Distance (SWD)[31] and Maximum Mean Discrepancy (MMD)[32] together with visualization as our criterion. The detailed setup of training and evaluation can be found in Appendix.C.

**Synthetic Datasets:** The Petal [9] and Gaussian Mixture Model (GMM) dataset are simple yet challenging, as they mimic natural dynamics arising in cellular differentiation, including bifurcations and merges. We compare our algorithm with MIOFlow in Fig.3. DMSB can infer trajectories aligned with ground truth distribution more faithfully at timesteps when snapshots are taken.

In GMM experiments (see Fig.4), we choose standard Gaussian at initial and terminal time steps while four-modal GMM and eight-modal GMM are placed at intermediate time steps. Besides good position trajectory, it is almost serendipity that DMSB can also learn the reasonable velocity trajectory *without* any access to ground truth velocity information. This paves the way for our later velocity estimation for the RNAsc dataset. **scRNA-seq Dataset:** The emergence of single-cell profiling technologies has facilitated the acquisition of high-resolution single-cell data, enabling the characterization of individual cells at distinct developmental states [7]. However, because the cell population is eliminated after the measurement, one may only gather statistical data for single samples at particular timesteps, which neither preserves

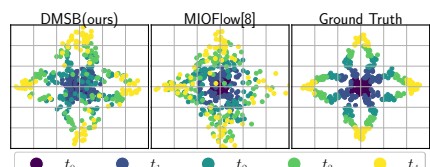

Figure 3: Comparsion with MIOFlow and ground truth on challenging petal dataset. DMSB is able to generate trajectories whose time marginal matches ground truth faithfully and outperforms prior work. Time is indicated by colors.

any correlations over time nor provides access to the ground truth trajectory. The diversity of embryonic stem cells after development from embryoid bodies, which comprises mesoderm, endoderm,

Table 3: Numerical result of MMD and SWD on 100 dimensions single-cell RNA-seq dataset and results for leaving out (LO) marginals at different observation. DMSB outperforms prior work by a large margin for both metrics and all leave-out case. See Appendix.4 for Results over 3 seeds.

| | MMD $\downarrow$ | | | | SWD $\downarrow$ | | | |
|---|---|---|---|---|---|---|---|---|
| Algorithm | w/o LO | LO-$t_1$ | LO-$t_2$ | LO-$t_3$ | w/o LO | LO-$t_1$ | LO-$t_2$ | LO-$t_3$ |
| NLSB[10] | 0.66 | 0.38 | 0.37 | 0.37 | 0.54 | 0.55 | 0.54 | 0.55 |
| MIOFlow[8] | 0.23 | 0.23 | 0.90 | 0.23 | 0.35 | 0.49 | 0.72 | 0.50 |
| DMSB(ours) | **0.03** | **0.04** | **0.04** | **0.04** | **0.20** | **0.20** | **0.19** | **0.18** |

neuroectoderm, and neural crest in 27 days, is demonstrated by the scRNA-seq dataset. The snapshot of cells are collected between ($t_0$: day 0 to 3, $t_1$: day 6 to 9, $t_2$: day 12 to 15, $t_3$: day 18 to 21,$t_4$: day 24 to 27). Snapshot data are prepossessed by the quality control [30] and then projected to feature space by principal component analysis (PCA). We inherit processed data from [8]. We validate DMSB on 5-dim and 100-dim PCA space to show superior performance on high-dimension problems compared with baselines. We further show that DMSB can estimate better velocity distribution compared with baselines when the ground truth is absent during training and testing.

We testify the performance of our model by computing MMD and SWD with full snapshots and when one of snapshots is left out (LO). We postpone the comparison of all the models on 5-d RNA space to the appendix (see Fig.9 and Table.6) because the problem is relatively simple and all models can infer accurate trajectory. Table.3 summarizes the average MMD and SWD between estimated marginal and ground truth over different snapshot timesteps. DMSB outperforms prior work by a large margin in high (100) dimensional scenarios. The visualization (Fig.5) in PCA space further justifies the numerical result and highlights the variety and quality of the samples produced by DMSB.

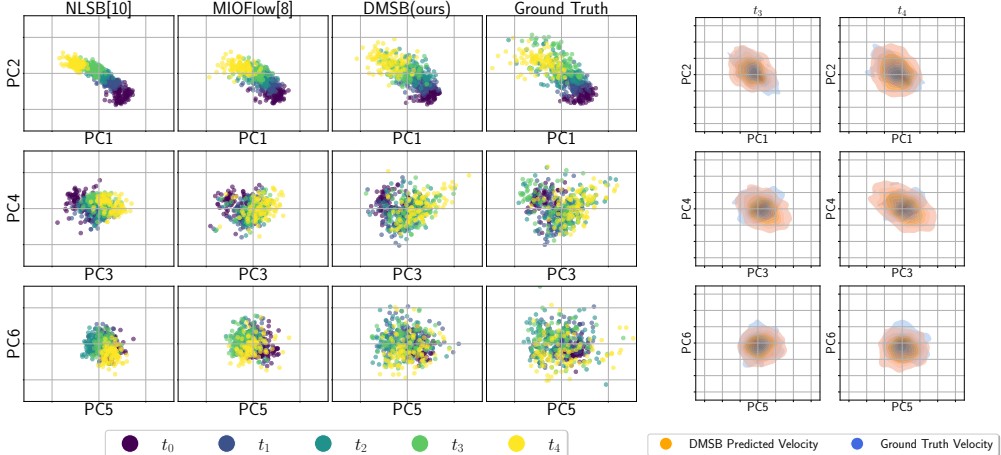

Figure 5: Comparison of population-level dynamics on 100-dimensional PCA space at the moment of observation for scRNA-seq data using MIOFlow, NLSB, and DMSB. We display the plot of the first 6 principle components (PC). Baselines can only learn the trajectory's fundamental trend, whereas DMSB can match the target marginal along the trajectory across different dimensions. The right figure shows Kernel Density Estimation [33] of samples generated by DMSB and ground truth at $t_3$ and $t_4$. The generated samples for all timesteps and comparison with baseline are in Appendix.F.

Interestingly, Fig.4 demonstrates that DMSB can reconstruct reasonable evolution of the velocity distribution which was not accessible to the algorithm. We further validate such property in 100-D RNAsc dataset. During the training and testing, all the models do not have access to the ground truth velocity. We run the experiments of 100-D and 5-D RNAsc datasets and average the discrepancy between ground truth velocity and estimated velocity over snapshot time. The numerical values are listed in the Table.7 and Table.6. The plot of velocity and position can be found in Fig.9 and Fig.10. The plot illustrates that while all models are capable of learning reasonable trajectories, only DMSB has the ability to estimate a plausible velocity distribution. This property holds even for

100-D RNA dataset (see Fig.5,11,12). This is notable, despite the velocity estimated by DMSB does not perfectly match the ground truth, because it should be noted that the proposed phase space SDE and the optimality of OT are artificial and may not necessarily represent the actual RNA evolution. Moreover, as individual evolutions cannot be tracked, possibilities such as {A→A, B→B} versus {A→B, B→A} can not be discerned, which renders exact velocity recovering almost impossible.

## 6 Conclusion and Limitations

In this paper, we propose DMSB, a scalable algorithm that learns the trajectory which fits the different marginal distributions over time. We extend the mean matching objective to phase space which enables efficient mSB computing. We propose a novel training scheme to fit the mean matching objective without violating BI which is the root of solving mmmSB problem. We demonstrate the superior result of DMSB compared with the existing algorithms.

A main limitation of this work is, the rate of convergence to the actual mmmSB has not been quantified after neural network approximations are introduced. Even though [15] theoretically analyzed the convergence of mean matching iteration, supporting its outstanding performance [14], the iteration still fails to converge to the actual SB [34] precisely due to practical neural network estimation errors accumulating over BI. However, recent work [35] shows the convergence of SB when training error exists. In addition, DMSB cannot simulate the process with death and birth of cells which can be potentially described as unbalanced optimal transport [36].

## 7 Acknowledgement

This research was supported by the ARO Award W911NF2010151, and the DoD Basic Research Office AwardHQ00342110002.

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

# A  Appendix

# B  Proof in §3 and §4

Before stating our proofs, we provide the assumptions used throughout the paper. These assumptions are adopted from stochastic analysis for SGM [27, 37, 38], SB [39], and FBSDE [40–42].

(i)  $\mu_{t_i}$ with finite second-order moment for all $t_i$.

(ii)  $\boldsymbol{f}$ and $g$ are continuous functions, and $|g(t)|^2 > 0$ is uniformly lower-bounded w.r.t. $t$.

(iii)  $\forall t \in [0, T]$, we have $\nabla_{\mathbf{v}} \log p_t(\mathbf{m}_t, t), \nabla_{\mathbf{v}} \log \Psi(\cdot, \cdot, \cdot), \nabla_{\mathbf{v}} \log \widehat{\Psi}(\cdot, \cdot, \cdot), \mathbf{Z}(\cdot, \cdot, \cdot; \theta)$, and $\widehat{\mathbf{Z}}(\cdot, \cdot, \cdot; \phi)$ Lipschitz and at most linear growth w.r.t. $\mathbf{x}$ and $\mathbf{v}$.

(iv)  $\Psi, \widehat{\Psi} \in C^{1,2}$.

(v)  $\exists k > 0 : p_t^{SB}(\mathbf{m}) = \mathcal{O}(\exp^{-\|\mathbf{m}\|_k^2})$ as $\mathbf{m} \to \infty$.

Assumptions (i) (ii) (iii) are standard conditions in stochastic analysis to ensure the existence-uniqueness of the SDEs; hence also appear in SGM analysis [37]. Assumption (iv) allows applications of Itô formula and properly defines the backward SDE in FBSDE theory. Finally, assumption (v) assures the exponential limiting behavior when performing integration by parts. w.o.l.g, we denote $\boldsymbol{f} = [\mathbf{v}, \mathbf{0}]^\mathsf{T}$.

## B.1  Proof of Proposition.3.1

The results of the Prop.3.1 is part of results of Prop.B.4 which gives the results for both forward and backward likelihood objective.

**Theorem B.1.** *The optimization problem*

$$\min \int_0^1 \int \frac{1}{2} \|\mathbf{a}\|_2^2 \mu \mathrm{d}\mathbf{x}\mathrm{d}\mathbf{v}\mathrm{d}t, \tag{12}$$

$$s.t \begin{cases} \frac{\partial \mu(\mathbf{m}_t)}{\partial t} = -\nabla_{\mathbf{m}} \cdot \{[(\boldsymbol{f} + g\mathbf{u})] \mu\} + \frac{1}{2}g^2 \Delta_{\mathbf{v}}\mu, \\ \mu_0 = p(0, \mathbf{x}, \mathbf{v}), \quad \mu_1 = p(T, \mathbf{x}, \mathbf{v}), \end{cases} \tag{13}$$

*will induce the coupled PDEs,*

$$\frac{\partial \mu(\mathbf{m}_t)}{\partial t} = -\nabla_{\mathbf{m}} \cdot \left[ \left( \boldsymbol{f} + g^2 \nabla_{\mathbf{v}}\phi \right) \mu \right] + \frac{1}{2}g^2 \Delta_{\mathbf{v}}\mu, \tag{14}$$

$$\frac{\partial \phi(\mathbf{m}_t)}{\partial t} = -\frac{1}{2}\|g\nabla_{\mathbf{v}}\phi\|_2^2 - \mathbf{v}^\mathsf{T}\nabla_{\mathbf{x}}\phi - \frac{1}{2}g^2 \Delta_{\mathbf{v}}\phi, \tag{15}$$

*and the optimal control of the problem is*

$$\mathbf{a}^* = g\nabla_{\mathbf{v}}\phi.$$

*Proof.* One can write the Lagrange by introducing lagrangian multiplier $\phi$:

$$
\mathcal{L}(\mu, \mathbf{a}, \phi) = \int_0^1 \int_{\mathbb{R}^n \times \mathbb{R}^n} \frac{1}{2} \|\mathbf{a}\|_2^2 \mu \mathrm{d}\mathbf{x} \mathrm{d}\mathbf{v} \mathrm{d}t + \int_0^1 \int_{\mathbb{R}^n \times \mathbb{R}^n} \phi \frac{\partial \mu}{\partial t} \mathrm{d}\mathbf{v} \mathrm{d}\mathbf{x} \mathrm{d}t
$$

$$
+ \int_0^1 \int_{\mathbb{R}^n \times \mathbb{R}^n} \phi \left\{ -\frac{1}{2} g^2 \Delta_{\mathbf{m}} \mu + \nabla_{\mathbf{m}} \cdot \left[ (\boldsymbol{f} + g\mathbf{u}) \mu \right] \right\} \mathrm{d}\mathbf{v} \mathrm{d}\mathbf{x} \mathrm{d}t
$$

$$
= \int_0^1 \int_{\mathbb{R}^n \times \mathbb{R}^n} \frac{1}{2} \|\mathbf{a}\|_2^2 \mu \mathrm{d}\mathbf{x} \mathrm{d}\mathbf{v} \mathrm{d}t - \int_0^1 \int_{\mathbb{R}^n \times \mathbb{R}^n} \mu \frac{\partial \phi}{\partial t} \mathrm{d}\mathbf{v} \mathrm{d}\mathbf{x} \mathrm{d}t
$$

$$
+ \int_0^1 \int_{\mathbb{R}^n \times \mathbb{R}^n} \phi \nabla_{\mathbf{m}} \cdot \left[ (\boldsymbol{f} + g\mathbf{u}) \mu \right] - \phi \left[ \frac{1}{2} g^2 \Delta_{\mathbf{m}} \mu \right] \mathrm{d}\mathbf{v} \mathrm{d}\mathbf{x} \mathrm{d}t
$$

$$
= \int_0^1 \int_{\mathbb{R}^n \times \mathbb{R}^n} \frac{1}{2} \|\mathbf{a}\|_2^2 \mu \mathrm{d}\mathbf{x} \mathrm{d}\mathbf{v} \mathrm{d}t - \int_0^1 \int_{\mathbb{R}^n \times \mathbb{R}^n} \mu \frac{\partial \phi}{\partial t} \mathrm{d}\mathbf{v} \mathrm{d}\mathbf{x} \mathrm{d}t
$$

$$
+ \int_0^1 \int_{\mathbb{R}^n \times \mathbb{R}^n} -\nabla_{\mathbf{m}} \phi^\mathsf{T} \left[ (\boldsymbol{f} + g\mathbf{u}) \right] \mu - \mu \left[ \frac{1}{2} g^2 \Delta_{\mathbf{m}} \phi \right] \mathrm{d}\mathbf{v} \mathrm{d}\mathbf{x} \mathrm{d}t
$$

$$
= \int_0^1 \int_{\mathbb{R}^n \times \mathbb{R}^n} \left\{ \frac{1}{2} \|\mathbf{a}\|_2^2 - \frac{\partial \phi}{\partial t} - \mathbf{v}^\mathsf{T} \nabla_{\mathbf{x}} \phi - \frac{1}{2} g^2 \Delta_{\mathbf{v}} \phi - g \nabla_{\mathbf{v}} \phi^\mathsf{T} \mathbf{a} \right\} \mu \ \mathrm{d}\mathbf{v} \mathrm{d}\mathbf{x} \mathrm{d}t
$$

By taking the minimization within the bracket, The optimal control is,

$$
\mathbf{a}^* = g \nabla_{\mathbf{v}} \phi
$$

By Plugging it back, the optimality of the aforementioned problem is presented as:

$$
\frac{\partial \mu(\mathbf{m}_t)}{\partial t} = -\nabla_{\mathbf{v}} \cdot \left[ \left( \boldsymbol{f} + g^2 \nabla_{\mathbf{v}} \phi \right) \mu \right] + \frac{1}{2} g^2 \Delta_{\mathbf{v}} \mu,
$$

$$
\frac{\partial \phi(\mathbf{m}_t)}{\partial t} = -\frac{1}{2} \| g \nabla_{\mathbf{v}} \phi \|_2^2 - \mathbf{v}^\mathsf{T} \nabla_{\mathbf{x}} \phi - \frac{1}{2} g^2 \Delta_{\mathbf{v}} \phi,
$$

$\square$

**Theorem B.2.** *The optimal forward and backward processes are represented as:*

$$
\mathrm{d}\mathbf{m}_t = \left[ \boldsymbol{f} + g\mathbf{u}_t^{f*} \right] \mathrm{d}t + g(t)\mathrm{d}\mathbf{w}_t \quad \textit{(forward)} \tag{16}
$$

$$
\mathrm{d}\mathbf{m}_s = \left[ \boldsymbol{f} + g\mathbf{u}_s^{b*} \right] \mathrm{d}t + g(t)\mathrm{d}\mathbf{w}_s \quad \textit{(Backward)} \tag{17}
$$

*in which* $\boldsymbol{f} = [\mathbf{v}, \mathbf{0}]^\mathsf{T}$. *Optimal control is expressed as,*

$$
\mathbf{u}_t^{f*} := \mathbf{Z}_t \equiv \begin{pmatrix} \mathbf{0} \\ \mathbf{z}_t \end{pmatrix} \equiv \begin{pmatrix} \mathbf{0} \\ g \nabla_{\mathbf{v}} \log \Psi_t \end{pmatrix} \tag{18}
$$

$$
\mathbf{u}_t^{b*} := \widehat{\mathbf{Z}}_t \equiv \begin{pmatrix} \mathbf{0} \\ \widehat{\mathbf{z}}_t \end{pmatrix} \equiv \begin{pmatrix} \mathbf{0} \\ g \nabla_{\mathbf{v}} \log \widehat{\Psi}_t \end{pmatrix} \tag{19}
$$

*where* $\Psi$ *and* $\widehat{\Psi}$ *are the solution of following PDEs,*

$$
\boxed{
\begin{aligned}
& \frac{\partial \Psi_t}{\partial t} = -\frac{1}{2} g^2 \Delta_{\mathbf{v}} \Psi_t - \nabla_{\mathbf{x}} \Psi_t^\mathsf{T} \mathbf{v} \\
& \frac{\partial \widehat{\Psi}_t}{\partial t} = \frac{1}{2} g^2 \Delta_{\mathbf{v}} \widehat{\Psi}_t - \nabla_{\mathbf{x}} \widehat{\Psi}_t^\mathsf{T} \mathbf{v} \\
& s.t \quad \Psi(\mathbf{x}, \mathbf{v}, 0) \widehat{\Psi}(\mathbf{x}, \mathbf{v}, 0) = p(\mathbf{x}, \mathbf{v}, 0), \quad \Psi(\mathbf{x}, \mathbf{v}, T) \widehat{\Psi}(\mathbf{x}, \mathbf{v}, T) = p(\mathbf{x}, \mathbf{v}, T)
\end{aligned}
} \tag{20}
$$

*Proof.* By Lemma.B.1, we notice that the optimal control is:

$$
\mathbf{a}^* = g \nabla_{\mathbf{v}} \phi.
$$

By leveraging Hopf-Cole [43, 44] transformation, here we define

$$
\Psi = \exp(\phi),
$$

$$
\widehat{\Psi} = \mu \exp(-\phi).
$$

Then we can have the following expressions:

$$\nabla \Psi = \exp(\phi)\nabla\phi$$
$$\Delta \Psi = \nabla \cdot (\nabla \Psi)$$
$$= \sum_i \frac{\partial}{\partial \mathbf{m}_i}\left[\exp\left(\phi\right)\nabla\phi\right]$$
$$= \left[\nabla\phi^{\mathsf{T}}\left(\exp\left(\phi\right)\frac{\partial\phi_i}{\partial\mathbf{m}_i}\right) + \exp\left(\phi\right)\frac{\partial(\nabla\phi)_i}{\partial\mathbf{m}_i}\right]$$
$$= \exp\left(\phi\right)\left[\|\nabla\phi\|_2^2 + \Delta\phi\right]$$
$$\nabla\widehat{\Psi} = \mu\exp\left(-\phi\right)\left(-\nabla\phi\right) + \exp(-\phi)\nabla\mu$$
$$= \exp(-\phi)(-\mu\nabla\phi + \nabla\mu)$$
$$\Delta\widehat{\Psi} = \nabla \cdot \left(\nabla\widehat{\Psi}\right)$$
$$= \sum_i \frac{\partial}{\partial\mathbf{m}_i}\left[\exp\left(-\phi\right)\left(-\mu\nabla\phi + \nabla\mu\right)\right]$$
$$= \sum_i \left[\left(-\mu\nabla\phi + \nabla\mu\right)^{\mathsf{T}}\left(\exp\left(-\phi\right)\frac{-\partial[\nabla\phi]_i}{\partial\mathbf{m}_i}\right)\right.$$
$$\left. + \exp\left(-\phi\right)\left(\frac{\partial}{\partial\mathbf{m}_i}[\nabla\mu]_i - \mu\frac{\partial}{\partial\mathbf{m}_i}[\nabla\phi]_i - \nabla\phi^{\mathsf{T}}\frac{\partial}{\partial\mathbf{m}_i}[\nabla\mu]_i\right)\right]$$
$$= \exp\left(-\phi\right)\left[\mu\|\nabla\phi\|_2^2 - \nabla\mu^{\mathsf{T}}\nabla\phi + \Delta\mu - \mu\Delta\phi - \nabla\phi^{\mathsf{T}}\nabla\mu\right]$$
$$= \exp\left(-\phi\right)\left[\mu\|\nabla\phi\|_2^2 - 2\nabla\mu^{\mathsf{T}}\nabla\phi + \Delta\mu - \mu\Delta\phi\right]$$

Thus, we can have.

$$\frac{\partial\Psi}{\partial t} = \exp\left(\phi\right)\frac{\partial\phi}{\partial t}$$
$$= \exp\left(\phi\right)\left(-\frac{1}{2}\|g\nabla_{\mathbf{v}}\phi\|_2^2 - \mathbf{v}^{\mathsf{T}}\nabla_{\mathbf{x}}\phi - \frac{1}{2}g^2\Delta_{\mathbf{v}}\phi\right)$$
$$= -\frac{1}{2}g^2\Delta_{\mathbf{v}}\Psi - \nabla_{\mathbf{x}}\Psi^{\mathsf{T}}\mathbf{v}$$
$$\frac{\partial\widehat{\Psi}}{\partial t} = \exp\left(-\phi\right)\frac{\partial\mu}{\partial t} - \mu\exp\left(-\phi\right)\frac{\partial\phi}{\partial t}$$
$$= \exp\left(-\phi\right)\left(\frac{\partial\mu}{\partial t} - \mu\frac{\partial\phi}{\partial t}\right)$$
$$= \exp\left(-\phi\right)\left[-\nabla_{\mathbf{m}}\cdot\{[(\boldsymbol{f} + g\mathbf{u})\,\mathbf{I}_d]\,\mu\} + \frac{1}{2}g^2\Delta_{\mathbf{m}}\mu + \mu\left(\frac{1}{2}\|g\nabla_{\mathbf{v}}\phi\|_2^2 + \mathbf{v}^{\mathsf{T}}\nabla_{\mathbf{x}}\phi + \frac{1}{2}g^2\Delta_{\mathbf{v}}\phi\right)\right]$$
$$= \exp\left(-\phi\right)\left[-\nabla_{\mathbf{v}}\cdot(g^2\mu\nabla_{\mathbf{v}}\phi) - \mathbf{v}^{\mathsf{T}}\nabla_{\mathbf{x}}\mu + \frac{1}{2}g^2\Delta_{\mathbf{v}}\mu + \frac{\mu}{2}\|g\nabla_{\mathbf{v}}\phi\|_2^2 + \mu\mathbf{v}^{\mathsf{T}}\nabla_{\mathbf{x}}\phi + \mu\frac{1}{2}g^2\Delta_{\mathbf{v}}\phi\right]$$
$$= \exp\left(-\phi\right)\left[-g^2\nabla_{\mathbf{v}}\mu^{\mathsf{T}}\nabla_{\mathbf{v}}\phi - g^2\mu\Delta_{\mathbf{v}}\phi - \mathbf{v}^{\mathsf{T}}\nabla_{\mathbf{x}}\mu + \frac{1}{2}g^2\Delta_{\mathbf{v}}\mu + \frac{\mu}{2}\|g\nabla_{\mathbf{v}}\phi\|_2^2 + \mu\mathbf{v}^{\mathsf{T}}\nabla_{\mathbf{x}}\phi + \mu\frac{1}{2}g^2\Delta_{\mathbf{v}}\phi\right]$$
$$= \exp\left(-\phi\right)\left[-g^2\nabla_{\mathbf{v}}\mu^{\mathsf{T}}\nabla_{\mathbf{v}}\phi - \mu\frac{1}{2}g^2\Delta_{\mathbf{v}}\phi - \mathbf{v}^{\mathsf{T}}\nabla_{\mathbf{x}}\mu + \frac{1}{2}g^2\Delta_{\mathbf{v}}\mu + \frac{\mu}{2}\|g\nabla_{\mathbf{v}}\phi\|_2^2 + \mu\mathbf{v}^{\mathsf{T}}\nabla_{\mathbf{x}}\phi\right]$$
$$= \exp\left(-\phi\right)\left[\frac{\mu}{2}\|g\nabla_{\mathbf{v}}\phi\|_2^2 - g^2\nabla_{\mathbf{v}}\mu^{\mathsf{T}}\nabla_{\mathbf{v}}\phi - \mu\frac{1}{2}g^2\Delta_{\mathbf{v}}\phi + \frac{1}{2}g^2\Delta_{\mathbf{v}}\mu - \mathbf{v}^{\mathsf{T}}\nabla_{\mathbf{x}}\mu + \mu\mathbf{v}^{\mathsf{T}}\nabla_{\mathbf{x}}\phi\right]$$
$$= \frac{1}{2}g^2\Delta_{\mathbf{v}}\hat{\Psi} - \nabla_{\mathbf{x}}\widehat{\Psi}^{\mathsf{T}}\mathbf{v}$$

Then we can represent the optimal control as:

$$\mathbf{u}_t^{f*} := \mathbf{Z}_t \equiv \begin{pmatrix} \mathbf{0} \\ \mathbf{z}_t \end{pmatrix} \tag{21}$$

$$\equiv \begin{pmatrix} \mathbf{0} \\ g\nabla_{\mathbf{v}}\phi \end{pmatrix} \overset{\text{Hopf-Cole}}{=} \begin{pmatrix} \mathbf{0} \\ g\nabla_{\mathbf{v}} \log \Psi_t \end{pmatrix} \tag{22}$$

Then the solution of such mSB is characterized by the forward SDE:

$$\mathbf{dm}_t = \left[ \boldsymbol{f} + g\mathbf{u}_t^{f*} \right] \mathrm{d}t + g(t)\mathbf{dw}_t, \tag{23}$$

Due to the structure of Hopf-Cole transform, one can have

$$p_t^{SB} = p_t^{eq.(23)} = \Psi_t \widehat{\Psi}_t \tag{24}$$

According to [27, 28], the reverse drift of such SDE (eq.23) $\mathbf{u}_t^{b*}$ should admits,

$$\mathbf{u}_t^{f*} + \mathbf{u}_t^{b*} = \mathbf{g}\nabla_{\mathbf{v}} \log p_t^{SB} \tag{25}$$

$$\begin{pmatrix} \mathbf{0} \\ g\nabla_{\mathbf{v}} \log \Psi_t \end{pmatrix} + \mathbf{u}_t^{b*} = \begin{pmatrix} \mathbf{0} \\ g\nabla_{\mathbf{v}} \log \Psi_t + g\nabla_{\mathbf{v}} \log \widehat{\Psi}_t \end{pmatrix} \tag{26}$$

$$\mathbf{u}_t^{b*} = \begin{pmatrix} \mathbf{0} \\ g\nabla_{\mathbf{v}} \log \widehat{\Psi}_t \end{pmatrix} \tag{27}$$

which yields The backward optimal control

$$\mathbf{u}_t^{b*} := \widehat{\mathbf{Z}}_t \equiv \begin{pmatrix} \mathbf{0} \\ \widehat{\mathbf{z}}_t \end{pmatrix} \equiv \begin{pmatrix} \mathbf{0} \\ g\nabla_{\mathbf{v}} \log \widehat{\Psi}_t \end{pmatrix} \tag{28}$$

Thus, the optimal forward and backward process is

$$\mathbf{dm}_t = \left[ \boldsymbol{f} + g\mathbf{u}_t^{f*} \right] \mathrm{d}t + g(t)\mathbf{dw}_t \tag{29}$$

$$\mathbf{dm}_s = \left[ \boldsymbol{f} + g\mathbf{u}_s^{b*} \right] \mathrm{d}t + g(t)\mathbf{d}\widehat{\mathbf{w}}_s \tag{30}$$

And $\Psi$ and $\widehat{\Psi}$ satisfy following PDEs,

$$\frac{\partial \Psi_t}{\partial t} = -\frac{1}{2}g^2 \Delta_{\mathbf{v}} \Psi_t - \nabla_{\mathbf{x}} \Psi_t^{\mathsf{T}} \mathbf{v}$$

$$\frac{\partial \widehat{\Psi}_t}{\partial t} = \frac{1}{2}g^2 \Delta_{\mathbf{v}} \widehat{\Psi}_t - \nabla_{\mathbf{x}} \widehat{\Psi}_t^{\mathsf{T}} \mathbf{v}$$

$\square$

**Lemma B.3.** *By specifying $\boldsymbol{f} = [\mathbf{v}, \mathbf{0}]^{\mathsf{T}}$, The PDE shown in 20 can be represented by following SDEs*

$$\begin{pmatrix} \mathrm{d}\mathbf{x} \\ \mathrm{d}\mathbf{v} \end{pmatrix} = \begin{pmatrix} \mathbf{v} \\ -g^2 \nabla_{\mathbf{v}} \log \Psi \end{pmatrix} \mathrm{d}t + \begin{pmatrix} \mathbf{0} & \mathbf{0} \\ \mathbf{0} & g \end{pmatrix} \mathrm{d}\mathbf{w} \tag{31}$$

$$\mathrm{d}\mathbf{y} = \frac{1}{2}\|\mathbf{z}\|^2 \mathrm{d}t + \mathbf{z}^{\mathsf{T}} \mathrm{d}\mathbf{w}_t \tag{32}$$

$$\mathrm{d}\widehat{\mathbf{y}} = \left[ \frac{1}{2}\|\widehat{\mathbf{z}}\|^2 + \mathbf{z}^{\mathsf{T}}\widehat{\mathbf{z}} + \nabla_{\mathbf{v}} \cdot g\widehat{\mathbf{z}} \right] \mathrm{d}t + \widehat{\mathbf{z}}^{\mathsf{T}} \mathrm{d}\mathbf{w}_t \tag{33}$$

$$\boldsymbol{s.t} : \exp\left( \mathbf{y}_0 + \widehat{\mathbf{y}}_0 \right) = p(\mathbf{x}, \mathbf{v}, 0), \quad \exp\left( \mathbf{y}_T + \widehat{\mathbf{y}}_T \right) = p(\mathbf{x}, \mathbf{v}, T) \tag{34}$$

*Where:*

$$\mathbf{y} \equiv \mathbf{y}(\mathbf{x}_t, \mathbf{v}, t) = \log \Psi(\mathbf{x}_t, \mathbf{v}_t, t), \quad \mathbf{z} \equiv \mathbf{z}(\mathbf{x}_t, \mathbf{v}_t, t) = g\nabla_{\mathbf{v}} \log \Psi(\mathbf{x}_t, \mathbf{v}_t, t)$$

$$\widehat{\mathbf{y}} \equiv \widehat{\mathbf{y}}(\mathbf{x}_t, \mathbf{v}_t, t) = \log \widehat{\Psi}(\mathbf{x}_t, \mathbf{v}_t, t), \quad \widehat{\mathbf{z}} \equiv \widehat{\mathbf{z}}(\mathbf{x}_t, \mathbf{v}_t, t) = g\nabla_{\mathbf{v}} \log \widehat{\Psi}(\mathbf{x}_t, \mathbf{v}_t, t)$$

*Proof.* One can write

$$\frac{\partial \log \Psi}{\partial t} = \frac{1}{\Psi}\left(-\nabla_{\mathbf{x}}\Psi^{\mathsf{T}}\mathbf{v} - \frac{1}{2}g^2\Delta_{\mathbf{v}}\Psi\right)$$

$$= -\nabla_{\mathbf{x}}\log\Psi^{\mathsf{T}}\mathbf{v} - \frac{1}{2}g^2\frac{\Delta_{\mathbf{v}}\Psi}{\Psi}$$

$$= -\nabla_{\mathbf{x}}\log\Psi^{\mathsf{T}}\mathbf{v} - \frac{1}{2}g^2\operatorname{Tr}\left[\frac{1}{\Psi}\nabla_{\mathbf{v}}^2\Psi\right]$$

$$\frac{\partial \log \widehat{\Psi}}{\partial t} = \frac{1}{\widehat{\Psi}}\left(-\nabla_{\mathbf{x}}\widehat{\Psi}^{\mathsf{T}}\mathbf{v} + \frac{1}{2}g^2\Delta_{\mathbf{v}}\widehat{\Psi}\right)$$

$$= -\nabla_{\mathbf{x}}\log\widehat{\Psi}^{\mathsf{T}}\mathbf{v} - \frac{1}{2}g^2\operatorname{Tr}\left[\frac{1}{\widehat{\Psi}}\nabla_{\mathbf{v}}^2\widehat{\Psi}\right]$$

By applying Itô's lemma,

$$d\log\Psi = \frac{\partial \log \Psi}{\partial t}dt + \left[\nabla_{\mathbf{x}}\log\Psi^{\mathsf{T}}\mathbf{v} + g^2\|\nabla_{\mathbf{v}}\log\Psi\|_2^2 + \frac{1}{2}g^2\Delta_{\mathbf{v}}\log\Psi\right]dt + \left[\nabla_{\mathbf{m}}\log\Psi^{\mathsf{T}}\right]g d\mathbf{w}_t$$

$$= \left[-\nabla_{\mathbf{x}}\log\Psi^{\mathsf{T}}\mathbf{v} - \frac{1}{2}g^2\operatorname{Tr}\left[\frac{1}{\Psi}\nabla_{\mathbf{v}}^2\Psi\right]\right]dt$$

$$+ \left[\nabla_{\mathbf{x}}\log\Psi^{\mathsf{T}}\mathbf{v} + g^2\|\nabla_{\mathbf{v}}\log\Psi\|_2^2 + \frac{1}{2}g^2\operatorname{Tr}\left[\frac{1}{\Psi}\nabla_{\mathbf{v}}^2\Psi - \frac{1}{\Psi^2}\nabla_{\mathbf{v}}\Psi\nabla_{\mathbf{v}}\Psi^{\mathsf{T}}\right]\right]dt + g\left[\nabla_{\mathbf{v}}\log\Psi^{\mathsf{T}}\right]d\mathbf{w}_t$$

$$= \left[g^2\|\nabla_{\mathbf{v}}\log\Psi\|_2^2 - \frac{1}{2}g^2\operatorname{Tr}\left[\frac{1}{\Psi^2}\nabla_{\mathbf{v}}\Psi\nabla_{\mathbf{v}}\Psi^{\mathsf{T}}\right]\right]dt + g\left[\nabla_{\mathbf{v}}\log\Psi^{\mathsf{T}}\right]d\mathbf{w}_t$$

$$= \left[\frac{1}{2}g^2\|\nabla_{\mathbf{v}}\log\Psi\|_2^2\right]dt + g\left[\nabla_{\mathbf{v}}\log\Psi^{\mathsf{T}}\right]d\mathbf{w}_t$$

Similarly, one can have,

$$d\log\widehat{\Psi} = \frac{\partial \log \widehat{\Psi}}{\partial t}dt + \left[\nabla_{\mathbf{x}}\log\widehat{\Psi}^{\mathsf{T}}\mathbf{v} + g^2\nabla_{\mathbf{v}}\log\Psi^{\mathsf{T}}\nabla_{\mathbf{v}}\log\widehat{\Psi} + \frac{1}{2}g^2\Delta_{\mathbf{v}}\log\widehat{\Psi}\right]dt + \left[\nabla_{\mathbf{m}}\log\widehat{\Psi}^{\mathsf{T}}\right]g d\mathbf{w}_t$$

$$= \left[-\nabla_{\mathbf{x}}\log\widehat{\Psi}^{\mathsf{T}}\mathbf{v} + \frac{1}{2}g^2\operatorname{Tr}\left[\frac{1}{\widehat{\Psi}}\nabla_{\mathbf{v}}^2\widehat{\Psi}\right]\right]dt$$

$$+ \left[\nabla_{\mathbf{x}}\log\widehat{\Psi}^{\mathsf{T}}\mathbf{v} + g^2\nabla_{\mathbf{v}}\log\Psi^{\mathsf{T}}\nabla_{\mathbf{v}}\log\widehat{\Psi} + \frac{1}{2}g^2\Delta_{\mathbf{v}}\log\widehat{\Psi}\right]dt + g\left[\nabla_{\mathbf{v}}\log\widehat{\Psi}^{\mathsf{T}}\right]d\mathbf{w}_t$$

Noticing:

$$\frac{1}{2}\left[\frac{1}{\widehat{\Psi}}\nabla_{\mathbf{v}}^2\widehat{\Psi} + \nabla_{\mathbf{v}}^2\log\widehat{\Psi}\right] = \operatorname{Tr}\left[\frac{1}{\Psi}\nabla_{\mathbf{v}}^2\widehat{\Psi} - \frac{1}{2}\|\nabla_{\mathbf{v}}\log\widehat{\Psi}\|^2\right]$$

$$= \frac{1}{2}\|\nabla_{\mathbf{v}}\log\widehat{\Psi}\|^2 + \Delta_{\mathbf{v}}\log\widehat{\Psi}$$

Following the above derivation, one can have,

$$d\log\widehat{\Psi} = \left[-\nabla_{\mathbf{x}}\log\widehat{\Psi}^{\mathsf{T}}\mathbf{v} + \frac{1}{2}g^2\operatorname{Tr}\left[\frac{1}{\widehat{\Psi}}\nabla_{\mathbf{v}}^2\widehat{\Psi}\right]\right]dt$$

$$+ \left[\nabla_{\mathbf{x}}\log\widehat{\Psi}^{\mathsf{T}}\mathbf{v} + g^2\nabla_{\mathbf{v}}\log\Psi^{\mathsf{T}}\nabla_{\mathbf{v}}\log\widehat{\Psi} + \frac{1}{2}g^2\Delta_{\mathbf{v}}\log\widehat{\Psi}\right]dt + g\left[\nabla_{\mathbf{v}}\log\widehat{\Psi}^{\mathsf{T}}\right]d\mathbf{w}_t$$

$$= \left[g^2\nabla_{\mathbf{v}}\log\Psi^{\mathsf{T}}\nabla_{\mathbf{v}}\log\widehat{\Psi} + \frac{1}{2}g^2\|\nabla_{\mathbf{v}}\log\widehat{\Psi}\|^2 + 2\frac{1}{2}g^2\Delta_{\mathbf{v}}\log\widehat{\Psi}\right]dt + g\left[\nabla_{\mathbf{v}}\log\widehat{\Psi}^{\mathsf{T}}\right]d\mathbf{w}_t$$

By defining

$$\mathbf{y} \equiv \mathbf{y}(\mathbf{x}_t, \mathbf{v}, t) = \log \Psi(\mathbf{x}_t, \mathbf{v}_t, t), \quad \mathbf{z} \equiv \mathbf{z}(\mathbf{x}_t, \mathbf{v}_t, t) = g\nabla_{\mathbf{v}} \log \Psi(\mathbf{x}_t, \mathbf{v}_t, t)$$
$$\widehat{\mathbf{y}} \equiv \widehat{\mathbf{y}}(\mathbf{x}_t, \mathbf{v}_t, t) = \log \widehat{\Psi}(\mathbf{x}_t, \mathbf{v}_t, t), \quad \widehat{\mathbf{z}} \equiv \widehat{\mathbf{z}}(\mathbf{x}_t, \mathbf{v}_t, t) = g\nabla_{\mathbf{v}} \log \widehat{\Psi}(\mathbf{x}_t, \mathbf{v}_t, t)$$

One can conclude the results.

$$\begin{pmatrix} \mathrm{d}\mathbf{x} \\ \mathrm{d}\mathbf{v} \end{pmatrix} = \begin{pmatrix} \mathbf{v} \\ -g^2\nabla_{\mathbf{v}} \log \Psi \end{pmatrix} \mathrm{d}t + \begin{pmatrix} \mathbf{0} & \mathbf{0} \\ \mathbf{0} & g \end{pmatrix} \mathrm{d}\mathbf{w}$$
$$\mathrm{d}\mathbf{y} = \frac{1}{2}\|\mathbf{z}\|^2\mathrm{d}t + \mathbf{z}^\mathsf{T}\mathrm{d}\mathbf{w}_t$$
$$\mathrm{d}\widehat{\mathbf{y}} = \left[ \frac{1}{2}\|\widehat{\mathbf{z}}\|^2 + \mathbf{z}^\mathsf{T}\widehat{\mathbf{z}} + \nabla_{\mathbf{v}} \cdot g\widehat{\mathbf{z}} \right] \mathrm{d}t + \widehat{\mathbf{z}}^\mathsf{T}\mathrm{d}\mathbf{w}_t$$
$$\mathbf{s.t} : \exp\left(\mathbf{y}_0 + \widehat{\mathbf{y}}_0\right) = p(\mathbf{x}, \mathbf{v}, 0), \quad \exp\left(\mathbf{y}_T + \widehat{\mathbf{y}}_T\right) = p(\mathbf{x}, \mathbf{v}, T)$$

$\square$

**Proposition B.4.** *The log-likelihood at data point* $\mathbf{m}_0$ *can be expressed as*

$$\log p(\mathbf{m}_0, 0) = \mathbb{E}_{\mathbf{m}_t \sim (17)} \left[\log p(\mathbf{m}_T, T)\right] - \int_0^T \mathbb{E}_{\mathbf{m}_t \sim (17)} \left[ \frac{1}{2}\|\mathbf{z}_t\|^2 \mathrm{d}t + \frac{1}{2}\|\widehat{\mathbf{z}}_t\|^2 + \mathbf{z}_t^\mathsf{T}\widehat{\mathbf{z}}_t + \nabla_{\mathbf{v}} \cdot g\widehat{\mathbf{z}}_t \right] \mathrm{d}t$$

$$= \mathbb{E}_{\mathbf{m}_t \sim (17)} \left[\log p(\mathbf{m}_T, T)\right] -$$

$$\int_0^T \mathbb{E}_{\mathbf{m}_t \sim (17)} \left[ \frac{1}{2}\|\mathbf{z}_t\|^2 + \underbrace{\frac{1}{2}\|\widehat{\mathbf{z}}_t - g\nabla_{\mathbf{v}} \log p^{(17)} + \mathbf{z}_t\|^2}_{\textit{mean matching objective}} - \frac{1}{2}\|g\nabla_{\mathbf{v}} \log p^{(17)} - \mathbf{z}_t\|^2 \right] \mathrm{d}t$$

$$\propto \int_0^T \mathbb{E}_{\mathbf{m}_t \sim (17)} \left[ \underbrace{\frac{1}{2}\|\widehat{\mathbf{z}}_t - g\nabla_{\mathbf{v}} \log p^{(17)} + \mathbf{z}_t\|^2}_{\textit{mean matching objective}} \right] \mathrm{d}t$$

$$\log p(\mathbf{m}_T, T) = \mathbb{E}_{\mathbf{m}_t \sim (16)} \left[\log p(\mathbf{m}_0, 0)\right] - \int_0^T \mathbb{E}_{\mathbf{m}_t \sim (16)} \left[ \frac{1}{2}\|\mathbf{z}_t\|^2 \mathrm{d}t + \frac{1}{2}\|\widehat{\mathbf{z}}_t\|^2 + \mathbf{z}_t^\mathsf{T}\widehat{\mathbf{z}}_t + \nabla_{\mathbf{v}} \cdot g\mathbf{z}_t \right] \mathrm{d}t$$

$$= \mathbb{E}_{\mathbf{m}_t \sim (16)} \left[\log p(\mathbf{m}_0, 0)\right] -$$

$$\int_0^T \mathbb{E}_{\mathbf{m}_t \sim (16)} \left[ \frac{1}{2}\|\widehat{\mathbf{z}}_t\|^2 + \underbrace{\frac{1}{2}\|\widehat{\mathbf{z}}_t - g\nabla_{\mathbf{v}} \log p^{(16)} + \mathbf{z}_t\|^2}_{\textit{mean matching objective}} - \frac{1}{2}\|g\nabla_{\mathbf{v}} \log p^{(16)} - \widehat{\mathbf{z}}_t\|^2 \right] \mathrm{d}t$$

$$\propto \mathbb{E}_{\mathbf{m}_t \sim (16)} \left[ \underbrace{\frac{1}{2}\|\widehat{\mathbf{z}}_t - g\nabla_{\mathbf{v}} \log p^{(16)} + \mathbf{z}_t\|^2}_{\textit{mean matching objective}} \right] \mathrm{d}t$$

*By maximizing the log-likelihood at time* $t = 0$ *then* $t = T$ *iteratively,* $(\mathbf{z}_t, \widehat{\mathbf{z}}_t)$ *will converge to the solution of phase space SB.*

*Proof.* from Lemma.B.3, one can have:

$$\log p(\mathbf{m}_0, 0) = \mathbb{E}\left[\mathbf{y}_0 + \widehat{\mathbf{y}}_0\right]$$

$$= \mathbb{E}\left[\mathbf{y}_T + \widehat{\mathbf{y}}_T\right] - \int_0^T \mathbb{E}\left[\frac{1}{2}\|\mathbf{z}_t\|^2 \mathrm{d}t + \frac{1}{2}\|\widehat{\mathbf{z}}_t\|^2 + \mathbf{z}_t^\mathsf{T}\widehat{\mathbf{z}}_t + \nabla_\mathbf{v}\cdot g\widehat{\mathbf{z}}_t\right]\mathrm{d}t$$

$$= \mathbb{E}\left[\log p(\mathbf{m}_T, T)\right] - \int_0^T \mathbb{E}\left[\frac{1}{2}\|\mathbf{z}_t\|^2 + \frac{1}{2}\|\widehat{\mathbf{z}}_t\|^2 + \mathbf{z}_t^\mathsf{T}\widehat{\mathbf{z}}_t + \nabla_\mathbf{v}\cdot g\widehat{\mathbf{z}}_t\right]\mathrm{d}t$$

$$= \mathbb{E}\left[\log p(\mathbf{m}_T, T)\right] - \int_0^T \mathbb{E}\left[\frac{1}{2}\|\mathbf{z}_t\|^2 + \frac{1}{2}\|\widehat{\mathbf{z}}_t\|^2 - \widehat{\mathbf{z}}_t^\mathsf{T}\left(g\nabla_\mathbf{v}\log p^{SB}\right) + \mathbf{z}_t^\mathsf{T}\widehat{\mathbf{z}}_t\right]\mathrm{d}t$$

$$= \mathbb{E}\left[\log p(\mathbf{m}_T, T)\right]$$
$$- \int_0^T \mathbb{E}\left[\frac{1}{2}\|\mathbf{z}_t\|^2 + \frac{1}{2}\|\widehat{\mathbf{z}}_t - g\nabla_\mathbf{v}\log p^{SB} + \mathbf{z}_t\|^2 - \frac{1}{2}\|g\nabla_\mathbf{v}\log p^{SB} - \mathbf{z}_t\|^2\right]\mathrm{d}t$$

A similar result can be obtained for $\log p(\mathbf{m}_T, T)$.

One can notice that the likelihood objective is a continuous time analog of the mean matching objective proposed in [15], and iterative optimization between $logp(\mathbf{m}_0, 0)$ and $\log p(\mathbf{m}_T, T)$ are the continuous analog of IPF. Hence, the convergence proof will keep valid (see Proposition 4 in [15]). $\qquad\square$

The equivalence of KL divergence optimization in IPF and likelihood optimization is widely analyzed in [14, 15, 18]. The objective function will eventually boil down to the mean matching objective shown in the above proposition.B.4.

**Proposition B.5** (Optimality w.r.t. $\mathcal{K}_{\text{boundary}}$). *. Given the reference path measure $\bar{\pi}$ driven by the policy $\widehat{\mathbf{z}}_t$ from boundary $\mu_{t_{i+1}}$ in the reverse time direction, the optimal path measure in the forward time direction of the following problem*

$$\min_\pi \mathcal{J}(\pi) := \sum_{i=0}^{N-1} KL\left(\pi_{t_i:t_{i+1}} | \bar{\pi}_{t_i:t_{i+1}}\right), \quad s.t \quad \pi \in \left\{\int \pi_{t_i:t_{i+1}}\mathrm{d}\mathbf{m}_{t_{i+1}} = \mu_{t_i}, \int \mu_{t_i}\mathrm{d}\mathbf{v}_{t_i} = \rho_{t_i}\right\}$$

*is :*
$$\pi^*_{t_i:t_{i+1}} = \frac{\rho_{t_i}\bar{\pi}_{t_i:t_{i+1}}}{\int \bar{\pi}_{t_i:t_{i+1}}\mathrm{d}\mathbf{m}_{t_{i+1}}\mathrm{d}\mathbf{v}_{t_i}}.$$

*When $\pi_{t_i:t_{i+1}} \equiv \pi^*_{t_i:t_{i+1}}$, the following equations need to hold $\forall t \in [t_i, t_{i+1}]$:*

$$\|\mathbf{z}_t + \widehat{\mathbf{z}}_t - g\nabla_\mathbf{v}\log\hat{p}_t\|_2^2 = 0, \tag{35a}$$
$$p_{t_i}(\mathbf{v}_{t_i}|\mathbf{x}_{t_i}) \equiv \hat{q}_{t_i}(\mathbf{v}_{t_i}|\mathbf{x}_{t_i}), \tag{35b}$$

*where $\hat{p}_t$ and $\hat{q}_t$ denote the marginal density and conditional velocity distribution of the reference path measure at time $t$ and $t_i$, respectively.*

*Proof.* Due to the similarity of optimization for $\mathcal{K}_{\text{boundary}}$, the close form solution of the next path measure is (see §4 in [1] for detail):

$$\pi^*_{t_i:t_{i+1}} = \frac{\rho_{t_i}\bar{\pi}_{t_i:t_{i+1}}}{\int \bar{\pi}_{t_i:t_{i+1}}\mathrm{d}\mathbf{m}_{t_{i+1}}\mathrm{d}\mathbf{v}_{t_i}}.$$

By denoting the transition kernel of parameterized SDE driven by backward policy $\widehat{\mathbf{z}}_t$ as $q(\cdot|\cdot)$, and the time range between $t_i$ and $t_{i+1}$ is discretized into $S$ interval by EM discretization. Then one can

get

$$\pi^*_{t_i:t_{i+1}}$$

$$= \frac{\rho_{t_i}\bar{\pi}_{t_i:t_{i+1}}}{\int \bar{\pi}_{t_i:t_{i+1}}\mathrm{d}\mathbf{m}_{t_{i+1}}\mathrm{d}\mathbf{v}_{t_i}}$$

$$= \frac{p_{t_i}(\mathbf{x}_{t_i})q_{t_i}(\mathbf{m}_{t_i}|\mathbf{m}_{t_i+\delta_t})\cdots q_{t_{i+1}-\delta_t}(\mathbf{m}_{t_{i+1}-\delta_t}|\mathbf{m}_{t_{i+1}})\mu_{t_{i+1}}(\mathbf{m}_{t_{i+1}})}{q_{t_i}(\mathbf{x}_{t_i})}$$

$$= \frac{p_{t_i}(\mathbf{x}_{t_i})q_{t_i}(\mathbf{x}_{t_i},\mathbf{v}_{t_i}|\mathbf{x}_{t_i+\delta_t},\mathbf{v}_{t_i+\delta_t})q_{t_i+\delta_t}(\mathbf{x}_{t_i+\delta_t},\mathbf{v}_{t_i+\delta_t})\cdots q_{t_{i+1}-\delta_t}(\mathbf{m}_{t_{i+1}-\delta_t}|\mathbf{m}_{t_{i+1}})\mu_{t_{i+1}}(\mathbf{m}_{t_{i+1}})}{q_{t_i}(\mathbf{x}_{t_i})q_{t_i+\delta_t}(\mathbf{x}_{t_i+\delta_t},\mathbf{v}_{t_i+\delta_t})}$$

$$= \frac{p_{t_i}(\mathbf{x}_{t_i})q_{t_i}(\mathbf{x}_{t_i},\mathbf{v}_{t_i},\mathbf{x}_{t_i+\delta_t},\mathbf{v}_{t_i+\delta_t})\cdots q_{t_{i+1}-\delta_t}(\mathbf{m}_{t_{i+1}-\delta_t}|\mathbf{m}_{t_{i+1}})\mu_{t_{i+1}}(\mathbf{m}_{t_{i+1}})}{q_{t_i}(\mathbf{x}_{t_i})q_{t_i+\delta_t}(\mathbf{x}_{t_i+\delta_t},\mathbf{v}_{t_i+\delta_t})}$$

$$= \frac{p_{t_i}(\mathbf{x}_{t_i})q_{t_i}(\mathbf{v}_{t_i},\mathbf{x}_{t_i+\delta_t},\mathbf{v}_{t_i+\delta_t}|\mathbf{x}_{t_i})\cdots q_{t_{i+1}-\delta_t}(\mathbf{m}_{t_{i+1}-\delta_t}|\mathbf{m}_{t_{i+1}})\mu_{t_{i+1}}(\mathbf{m}_{t_{i+1}})}{q_{t_i+\delta_t}(\mathbf{m}_{t_i+\delta_t})}$$

$$= p_{t_i}(\mathbf{x}_{t_i})q_{t_i}(\mathbf{v}_{t_i},\mathbf{x}_{t_i+\delta_t},\mathbf{v}_{t_i+\delta_t}|\mathbf{x}_{t_i})\frac{q_{t_i+\delta_t}(\mathbf{m}_{t_i+\delta_t}|\mathbf{m}_{t_i+2\delta_t})\cdots q_{t_{i+1}-\delta_t}(\mathbf{m}_{t_{i+1}-\delta_t}|\mathbf{m}_{t_{i+1}})\mu_{t_{i+1}}(\mathbf{m}_{t_{i+1}})}{q_{t_i+\delta_t}(\mathbf{m}_{t_i+\delta_t})}$$

$$= p_{t_i}(\mathbf{x}_{t_i})q(\mathbf{v}_{t_i}|\mathbf{x}_{t_i})q(\mathbf{m}_{t_i+\delta_t}|\mathbf{m}_{t_i})\frac{q_{t_i+\delta_t}(\mathbf{m}_{t_i+\delta_t}|\mathbf{m}_{t_i+2\delta_t})\cdots q_{t_{i+1}-\delta_t}(\mathbf{m}_{t_{i+1}-\delta_t}|\mathbf{m}_{t_{i+1}})\mu_{t_{i+1}}(\mathbf{m}_{t_{i+1}})}{q_{t_i+\delta_t}(\mathbf{m}_{t_i+\delta_t})}$$

$$= p_{t_i}(\mathbf{x}_{t_i})q(\mathbf{v}_{t_i}|\mathbf{x}_{t_i})q(\mathbf{m}_{t_i+\delta_t}|\mathbf{m}_{t_i})\frac{q_{t_i+\delta_t}(\mathbf{m}_{t_i+2\delta_t}|\mathbf{m}_{t_i+\delta_t})\cdots q_{t_{i+1}-\delta_t}(\mathbf{m}_{t_{i+1}-\delta_t}|\mathbf{m}_{t_{i+1}})\mu_{t_{i+1}}(\mathbf{m}_{t_{i+1}})}{q_{t_i+\delta_t}(\mathbf{m}_{t_i+2\delta_t})} \tag{36}$$

**Doing eq.36 revursively**

$$= p_{t_i}(\mathbf{x}_{t_i})q(\mathbf{v}_{t_i}|\mathbf{x}_{t_i})q(\mathbf{m}_{t_i+\delta_t}|\mathbf{m}_{t_i})\prod_{s=1}^{S-1}q_s(\mathbf{m}_{t_i+(s+1)\cdot\delta_t}|\mathbf{m}_{t_i+s\cdot\delta_t})$$

$$= p_{t_i}(\mathbf{x}_{t_i})q(\mathbf{v}_{t_i}|\mathbf{x}_{t_i})\prod_{s=0}^{S-1}q_s(\mathbf{m}_{t_i+(s+1)\cdot\delta_t}|\mathbf{m}_{t_i+s\cdot\delta_t})$$

According to [15], given the policy $\hat{\mathbf{z}}_t$, the transition kernel $q_s(\mathbf{m}_{t_i+(s+1)\cdot\delta_t}|\mathbf{m}_{t_i+s\cdot\delta_t})$ can be estimated by $\hat{\mathbf{z}}_t$ (see Proposition 3 in [15])and it can be treated as the label for the forward policy $\mathbf{z}_t$ for all $s$. Thus, if $\pi_{t_it_{i+1}}$ is aligned with $\pi^*_{t_it_{i+1}}$, then one can construct following objective function for policy $\mathbf{z}_t$:

$$\mathcal{L} = \sum_t \|\underbrace{\mathbf{m}_t + \delta_t\mathbf{Z}_t(\mathbf{m}_t)}_{①} - (\mathbf{m}_t + \underbrace{\mathbf{m}_{t+\delta_t} + \delta_t\widehat{\mathbf{Z}}_{t+\delta_t}(\mathbf{m}_{t+\delta_t})}_{②} - \underbrace{(\mathbf{m}_t + \delta_t\widehat{\mathbf{Z}}_{t+\delta_t}(\mathbf{m}_t))}_{③}))\|_2^2 \tag{37}$$

$$= \sum_t \|\delta_t\mathbf{Z}_t(\mathbf{m}_t) + \delta_t\widehat{\mathbf{Z}}_{t+\delta_t}(\mathbf{m}_t) - (\mathbf{m}_{t+\delta_t} - \mathbf{m}_t - \delta_t\widehat{\mathbf{Z}}_{t+\delta_t}(\mathbf{m}_{t+\delta_t}))\|_2^2 \tag{38}$$

$$\approx \sum_t \|\mathbf{Z}_t(\mathbf{m}_t) + \widehat{\mathbf{Z}}_{t+\delta_t}(\mathbf{m}_t) - \nabla_\mathbf{v}\log p_t^{(17)}\|_2^2 \tag{39}$$

**due to the special structure of $\mathbf{Z}_t$ and $\widehat{\mathbf{Z}}_t$** $\tag{40}$

$$= \sum_t \|\mathbf{z}_t(\mathbf{m}_t) + \widehat{\mathbf{z}}_{t+\delta_t}(\mathbf{m}_t) - \nabla_\mathbf{v}\log p_t^{(17)}\|_2^2 \tag{41}$$

Where ①,②,③ corresponds to $F_k$, $B_k$, and $B_{k+1}$ in [15] respectively. Furthermore, we need to find a density function $p_{t_i}(\mathbf{v}_{t_i}|\mathbf{x}_{t_i})$ which satisfies

$$p_{t_i}(\mathbf{x}_{t_i})p_{t_i}(\mathbf{v}_{t_i}|\mathbf{x}_{t_i}) \equiv p_{t_i}(\mathbf{x}_{t_i})\hat{q}(\mathbf{v}_{t_i}|\mathbf{x}_{t_i})$$
$$p_{t_i}(\mathbf{v}_{t_i}|\mathbf{x}_{t_i}) \equiv \hat{q}_{t_i}(\mathbf{v}_{t_i}|\mathbf{x}_{t_i})$$

to be the new boundary condition. □

**Proposition B.6** (Optimality w.r.t. $\mathcal{K}_{\text{bridge}}$)**.** *Given the reference path measure $\bar{\pi}$ driven by the policy $\widehat{\mathbf{z}}_t$ from boundary $\mu_{t_N}$ in the reverse time direction, the optimal path measure in the forward time direction of the following problem*

$$\min_{\pi} \mathcal{J}(\pi) := \sum_{i=0}^{N-1} KL\left(\pi_{t_i:t_{i+1}} | \bar{\pi}_{t_i:t_{i+1}}\right), \quad s.t \quad \pi \in \mathcal{K}_{bridge} = \left\{\cap_{i=1}^{N-1} \mathcal{K}_{t_i}^2\right\}$$
$$is: \quad \pi_{t_0 t_N}^* = \frac{q_{t_0} \bar{\pi}_{t_0:t_N}}{\int \bar{\pi}_{t_0:t_N} \mathrm{d}\mathbf{m}_{t_N} \mathrm{d}\mathbf{v}_{t_0}}.$$

*when $\pi_{t_0 t_N} \equiv \pi_{t_0 t_N}^*$, the following equations need to hold $\forall t \in [t_0, t_N]$:*

$$\|\mathbf{z}_t + \widehat{\mathbf{z}}_t - g\nabla_{\mathbf{v}} \log \hat{p}_t\|_2^2 = 0 \tag{42a}$$

$$p_{t_0}(\mathbf{v}_{t_0}, \mathbf{x}_{t_0}) \equiv \hat{q}_{t_0}(\mathbf{v}_{t_0}, \mathbf{x}_{t_0}) \tag{42b}$$

*Proof.* Same proof as B.5. □

**Remark B.7.** The optimizer of such a problem can be represented as

$$\pi^* = \mu_{t_N} \bar{\pi}_{\cdot | t_N} \tag{43}$$

which can also be represented as,

$$\pi^* = \int \bar{\pi} \mathrm{d}\mu_{t_N} \cdot (\bar{\pi}_{\cdot | t_N})^R \tag{44}$$

Where the notation $R$ represents for the time reversal. The Proposition.4.4 is basically using neural network $\mathbf{Z}_t^\theta$ to approximate eq.44.

## C   Experiment Details

We test DMSB on 2D synthetic datasets and realworld scRNA-seq dataset [30]. We parameterize $\mathbf{z}(t, \mathbf{m}; \theta)$ and $\widehat{\mathbf{z}}(t, \mathbf{m}; \phi)$ with residual-based networks for all datasets (see.fig.6). The network adopts position encoding and is trained with AdamW[45] on one Nvidia 3090 Ti GPU. We use constant g(t) for simplicity though the framework can adopt time varying function g(t). We set the time horizon $T = t_N = 1 \cdot N$ and interval $\delta_t = 0.01$. We use EM discretization throughout the whole paper. For scRNA-seq dataset, we split data into train and test subsets(85% and 15%).All the experiment results are simulated by all-step push forward from initial data points at time $t = t_0$.

**MIOFlow and NLSB setup:** We use the official implementation of NLSB and MIOFlow.For MIOFlow, we report the best performance for all experiments w/GAE(or AE) and w/o GAE(or AE) embedding. For NLSB, we enlarge the size of the neural network to the best of our GPU capacity for a 100-dimensional scRNA-seq dataset and report the best performance during the training.

We evaluate the velocity of NLSB, as an SDE model, by its estimated drift term at time steps $t = \{1, 2, 3, 4, 5\}$. Because MIOflow w/ GAE simulates trajectories in the latent space, we estimate the velocity by using the forward finite difference technique with discretization $1E - 3$ sec after mapping from the latent code to the original space. We run the experiments of 100-D and 5-D RNAsc datasets and average the discrepancy between ground truth velocity and estimated velocity over snapshot time. The numerical values are listed in the Table.7 and Table.6. The plot of velocity and position can be found in Fig.9 and Fig.10. **We do not want to underestimate any prior work and tried out best to tune the prior work. Feel free to communicate with the first author if one can reproduce better results in the experiment section, and we are willing to update it.**

**Metrics and Evaluations** The 1-Wasserstein Distance suffers from the curse of dimensionality seriously. In the main paper, we are using Sliced-Wasserstein Distance (SWD) and Maximum Mean Distance as our criterion for 100-dim RNA dataset. An example is listed in the following toy code. One can notice that $W_1$ suffers from the curse of dimensionality seriously, the distance between two gaussian samples is even larger than the distance between gaussian and zeros (See following code snapshot). Hence such a metric is not suitable for high dimension ($\geq 100$) dataset evaluation even though some papers report $W_1$. In order to better evaluate our model compared with baselines, we are using $W_1$, Energy Distance, Max-sliced Wasserstein distance, Sliced-Wasserstein Distance and MMD. Our metric is adapted from Geoloss ($W_1$ and $Energy$), POT (Sliced Wassersetein and Maximum-Sliced Wasserstein) and this repo (MMD).

**Trajectories Cache** Similar to prior work [14, 15], we also need to cache the trajectories for training purposes. We cache 4096 trajectories for each Bregman Projection.

**Special Clarification for NLSB**

We evaluate the velocity of NLSB, as an SDE model, by its estimated drift term at time steps $t = \{1, 2, 3, 4, 5\}$. It may not be reasonable to consider the drift term as the real velocity, but the drift term can certainly depict a trend of SDE, so we still provide the result here.

```python
from ot.sliced import sliced_wasserstein_distance
a=torch.randn(1000,100) #1000 gaussian samples with dimension 100
b=torch.zeros(1000,100) #1000 zeros samples with dimension 100
c=torch.randn(1000,100) #1000 gaussian samples with dimension 100
Loss=sliced_wasserstein_distance
print('SWD distance between a and b is: {}'.format(Loss(a,b)))
print('SWD distance between a and c is: {}'.format(Loss(a,c)))
#SWD distance between a and b is: 1.0433608293533325
#SWD distance between a and c is: 0.11096614599227905
```
Listing 1: Distance compute by SWD distance with 1000 samples and 100 dimensions.

```python
from geomloss import SamplesLoss
a=torch.randn(1000,100) #1000 gaussian samples with dimension 100
b=torch.zeros(1000,100) #1000 zeros samples with dimension 100
c=torch.randn(1000,100) #1000 gaussian samples with dimension 100
Loss=SamplesLoss('sinkhorn',p=1)
print('W1 distance between a and b is: {}'.format(Loss(a,b)))
print('W1 distance between a and c is: {}'.format(Loss(a,c)))
#W1 distance between a and b is: 9.781818389892578
#W1 distance between a and c is: 11.734640121459961
```
Listing 2: Distance compute by $W_1$ distance with 1000 samples and 100 dimensions.

**Training:** We use Exponential Moving Average (EMA) with a decay rate of 0.999. Table.7 details the hyperparameters used for each dataset. The learning rate for all the datasets is set to be 2e-4 and the training batching size is 256. For computation efficiency, we cache large batch size of empirical samples from reference trajectory and sample training batch size from the cache data. The hyperparameters can be found in Table.7.

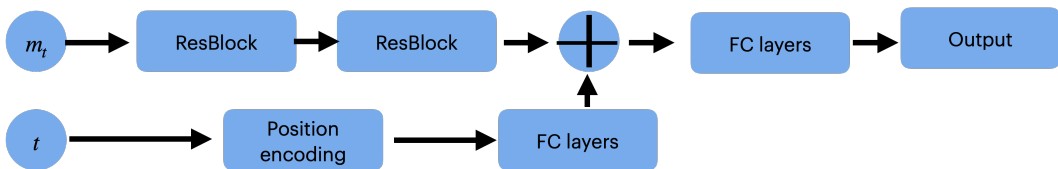

Figure 6: Neural network architecture for all experiments. The network size (# parameters) are varying between different tasks.

Figure 7: Training Hyper-parameters

| Dataset | time steps | # BI | $g(t)$ | # Parameters | $T$ | $SNR$ | # $\mathbf{v}_t$ Langevin |
|---|---|---|---|---|---|---|---|
| Semicircle | 15 | 2000 | 0.2 | 1.21M | 3 | 0.15 | 1 |
| Petal | 30 | 2000 | 0.2 | 1.21M | 2 | 0.15 | 1 |
| GMM | 15 | 2000 | 0.2 | 1.21M | 4 | 0.15 | 1 |
| scRNA (100 dim) | 15 | 4000 | 0.4 | 1.34M | 4 | 0.15 | 1 |

**Langevin sampling:** The Langevin sampling procedure for the velocity is summarized in 2. Given some pre-defined signal-to-noise ratio r (we set snr =0.15 for all experiments), the Langevin noise scale $\sigma$ at each time step t and each corrector step i is computed by

$$\sigma_t = \frac{2r^2 g^2 \|\epsilon\|^2}{\|\mathbf{z}(t, \mathbf{m}_t) + \widehat{\mathbf{z}}(t, \mathbf{m}_t)\|^2}, \tag{45}$$

# D Algorithms

---

**Algorithm 1** Sampling Procedure of DMSB

---

**Input:** Policies $\mathbf{z}(\cdot, \cdot; \theta)$ and $\widehat{\mathbf{z}}(\cdot, \cdot; \phi)$ Total sampling step $S = \frac{t_N}{\delta_t}$. Data distributions $\rho_{t_i}$. Initializing velocity distributions $\gamma_{t_i} = \mathcal{N}(0, \mathbf{I})$ if they are not avaliable.
**for** $s = 0$ **to** $S - 1$ **do**
    **if** s==0 **then**
        Sample position data $\mathbf{x}_{t_0}$ from $\rho_{t_0}$.
        **if** ground truth velocity distribution $\gamma_{t_0}$ avaliable **then**
            Sample velocity data $\mathbf{v}_{t_0}$ from $\gamma_{t_0}$
        **else**
            Sample velocity data $\mathbf{v}_{t_0}$ by Langevin simulation conditioning on $\mathbf{x}_{t_0}$.(Algorithm.2)
        **end if**
        $\mathbf{m}_{t_0} = [\mathbf{x}_{t_0}, \mathbf{v}_{t_0}]^\mathsf{T}$
    **end if**
    Simulating dynamics: $\mathrm{d}\mathbf{m}_t = [\boldsymbol{f}(\mathbf{m}_t, t) + g(t)\mathbf{Z}_t]\,\mathrm{d}t + g(t)\mathrm{d}\mathbf{w}_t$(eq.16)
**end for**
**return** $\mathbf{m}_{t \in [t_0, t_N]}$

---

---

**Algorithm 2** Langevin Sampler at $t_i$ marginal constraint

---

**Input:** policies $\mathbf{z}(\cdot, \cdot; \theta)$ and $\widehat{\mathbf{z}}(\cdot, \cdot; \phi)$, Previous timestep predicted velocity $\mathbf{v}_{t_i}$.
Sample position from ground truth $\mathbf{x}_{t_i} \sim \rho_{t_i}$.
**for** $step = 0$ **to** # Langevin steps **do**
    Sample $\epsilon \sim \mathcal{N}(\mathbf{0}, \boldsymbol{I})$.
    Construct new $\mathbf{m}_{t_i} = [\mathbf{x}_{t_i}, \mathbf{v}_{t_i}]^\mathsf{T}$
    Compute $\nabla_{\mathbf{v}} \log \tilde{p}_t^{\theta, \phi} \approx [\mathbf{z}(t_i, \mathbf{m}_{t_i}) + \widehat{\mathbf{z}}(t_i, \mathbf{m}_{t_i})]/g$.
    Compute $\sigma_t$ with (45).
    Langevin Sampling $\mathbf{v}_{t_i} \leftarrow \mathbf{v}_{t_i} + \sigma_{t_i} \nabla_{\mathbf{v}} \log \tilde{p}_{t_i}^{\theta, \phi} + \sqrt{2\sigma_t}\,\epsilon$.
**end for**
**return** $\mathbf{m}_{t_i} = [\mathbf{x}_{t_i}, \mathbf{v}_{t_i}]^\mathsf{T}$

---

---

**Algorithm 3** DMSB Training

---

**Input:** $N + 1$ Marginal position distribution $\rho_{t_i}, i \in [0, N]$.Parametrized policies $\mathbf{z}(\cdots ; \theta)$ and $\widehat{\mathbf{z}}(\cdots ; \phi)$. The number of Bregman Iteration $B$. Initialize postion and velocity at time step $t_i : \bar{\mathbf{m}}_{t_i} := None$ for the first iteration.

**if** Use ground truth velocity **then**
    set prior velocity: $\gamma_{t_i} = \gamma_{t_i}$
**else**
    set initial velocity $\gamma_{t_i} = \mathcal{N}(0, \boldsymbol{I})$
**end if**
**for** $b = 0$ **to** $B - 1$ **do**
    **for** $k = N$ **to** $1$ **do**
        $\mathbf{z}^{\phi}, \_ = OptSubSet(t_k, t_{k-1}, \mathbf{z}_{ref} = \mathbf{z}^{\theta}, \mathbf{z}_{opt} = \mathbf{z}^{\phi}, \eta = \phi, \bar{\mathbf{m}} = None)$ [Optimize $\mathcal{K}_{\text{boundary}}$]
    **end for**
    **for** $k = 0$ **to** $N - 1$ **do**
        $\mathbf{z}^{\theta}, \_ = OptSubSet(t_k, t_{k+1}, \mathbf{z}_{ref} = \mathbf{z}^{\phi}, \mathbf{z}_{opt} = \mathbf{z}^{\theta}, \eta = \theta, \bar{\mathbf{m}} = None)$ [Optimize $\mathcal{K}_{\text{boundary}}$]
    **end for**
    $\mathbf{z}^{\phi}, \widehat{\mathbf{m}} = OptSubSet(t_N, t_0, \mathbf{z}_{ref} = \mathbf{z}^{\theta}, \mathbf{z}_{opt} = \mathbf{z}^{\phi}, \eta = \phi, \bar{\mathbf{m}} = \bar{\mathbf{m}})$ [Optimize $\mathcal{K}_{\text{bridge}}$]
    **for** $k = 0$ **to** $N - 1$ **do**
        $\mathbf{z}^{\theta}, \_ = OptSubSet(t_k, t_{k+1}, \mathbf{z}_{ref} = \mathbf{z}^{\phi}, \mathbf{z}_{opt} = \mathbf{z}^{\theta}, \eta = \theta, \bar{\mathbf{m}} = \bar{\mathbf{m}})$ [Optimize $\mathcal{K}_{\text{boundary}}$]
    **end for**
    **for** $k = N$ **to** $1$ **do**
        $\mathbf{z}^{\phi}, \_ = OptSubSet(t_k, t_{k-1}, \mathbf{z}_{ref} = \mathbf{z}^{\theta}, \mathbf{z}_{opt} = \mathbf{z}^{\phi}, \eta = \phi, \bar{\mathbf{m}} = None)$ [Optimize $\mathcal{K}_{\text{boundary}}$]
    **end for**
    $\mathbf{z}^{\phi}, \widehat{\mathbf{m}} = OptSubSet(t_0, t_N, \mathbf{z}_{ref} = \mathbf{z}^{\phi}, \mathbf{z}_{opt} = \mathbf{z}^{\theta}, \eta = \theta, \bar{\mathbf{m}} = \bar{\mathbf{m}})$ [Optimize $\mathcal{K}_{\text{bridge}}$]
**end for**

---

**Algorithm 4** **Function** OptSubSet (Optimization for subsets)

---

**input:** Initial time $t_i$ and terminal time $t_j$. Reference path measure boundary condition $\rho_{t_i}$. Reference path measure driver $\mathbf{z}_{ref}$. Policy being optimized $\mathbf{z}_{opt}$ and corresponding parameter $\eta$. Empirical sample form last iteration $\widehat{\mathbf{m}}$.

**output:** $\mathbf{z}_{opt}$,samples $\widehat{\mathbf{m}}_{t_j}$ from reference path measure.

**if** $\widehat{\mathbf{m}}$ is None **then**
    Sample position data $\mathbf{x}_{t_i}$ from $\rho_{t_i}$.
    **if** velocity distribution $\gamma_{t_i}$ avaliable **then**
    Sample conditional velocity data $\mathbf{v}_{t_i}$ from $\gamma_{t_i}$
    **else**
    Sample velocity data $\mathbf{v}_{t_i}$ by Langevin simulation conditioning on $\mathbf{x}_{t_i}$.(see Algorithm.2.)
    **end if**
    $\mathbf{m}_{t_i} = [\mathbf{x}_{t_i}, \mathbf{v}_{t_i}]^{\mathsf{T}}$
**else**
    $\mathbf{m}_{t_i} = \bar{\mathbf{m}}$
**end if**
Sample trajectory $\mathbf{m}_{t \in [t_i, t_j]}$ from $\mathbf{m}_{t_i}$ using $\mathbf{z}_{ref}$
Compute $\mathcal{L} = \alpha \mathcal{L}_{\text{MM}} + (1 - \alpha)\mathcal{L}_{reg}$    (Regularization of SB $L_{reg}$[29] is optional )
update $\eta$

---

# E  Additional Diagram

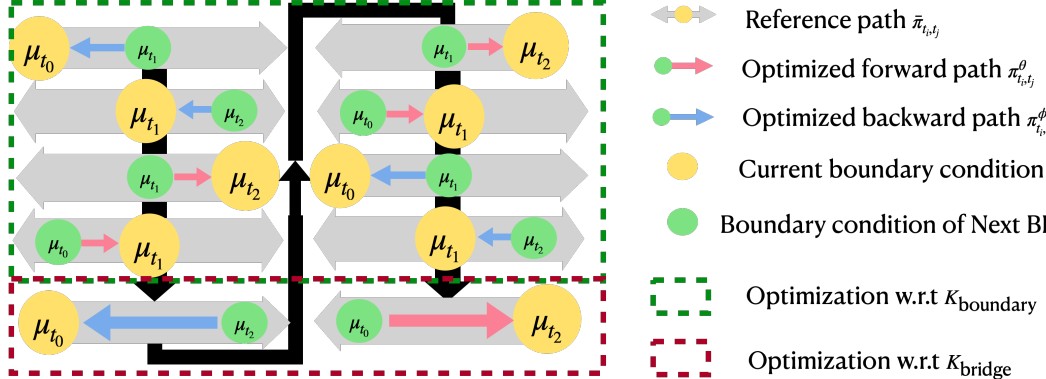

Figure 8: The detailed example diagram of Fig.2. We demonstrate an example of 3 marginals case. The training scheme can be extended to general N marginals easily. The figure consists of two BIs that differs by the training order. Given the reference path measure, we first run the Bregman Projection (BP) within the subset of $\mathcal{K}_{\text{boundary}}$ sequentially and end up with the constraint $\mathcal{K}_{\text{bridge}}$.

# F  Additional Experiment

Table 4: Our algorithm results over 3 seeds. Numerical result of MMD and SWD on 100 dimensions single-cell RNA-seq dataset and results for leaving out marginals at different observation. DMSB outperforms prior work by a large margin for both metrics and all leave-out case.

| LO | Metrics | $t_1$ | $t_2$ | $t_3$ | $t_4$ | Avg |
|---|---|---|---|---|---|---|
| w/o LO | MMD↓ | 0.021±1E-3 | 0.029±5E-3 | 0.038±2E-3 | 0.034±2E-3 | 0.032±3E-3 |
|  | SWD↓ | 0.114±5E-2 | 0.155±2E-2 | 0.19±3E-2 | 0.155±1E-2 | 0.16±2E-2 |
| w/ LO-$t_1$ | MMD↓ | 0.09±1E-3 | 0.019±1E-2 | 0.032±2E-2 | 0.029±2E-2 | 0.042±2E-2 |
|  | SWD↓ | 0.140±2E-2 | 0.155±1E-2 | 0.19±2E-2 | 0.155±1E-2 | 0.153±3E-2 |
| w/ LO-$t_2$ | MMD↓ | 0.021±1E-3 | 0.065±5E-3 | 0.032±2E-3 | 0.02±2E-3 | 0.033±3E-3 |
|  | SWD↓ | 0.100±5E-2 | 0.202±2E-2 | 0.13±3E-2 | 0.191±1E-2 | 0.155±2E-2 |
| w/ LO-$t_3$ | MMD↓ | 0.025±2E-3 | 0.026±2E-2 | 0.075±1E-2 | 0.029±2E-2 | 0.040±2E-2 |
|  | SWD↓ | 0.124±2E-2 | 0.14±1E-2 | 0.27±2E-2 | 0.18±1E-2 | 0.179±3E-2 |

Table 5: Numerical result of Wasserstein-1 ($W_1$), MMD, energy distance and Max-sliced Wasserstein distance (MWD) on **position** of 5 dimensions single-cell RNA-seq dataset using 500 generative samples and 500 ground truth data.

| Dim=5 | Energy ↓ | MMD ↓ | $W_1$ ↓ | SWD↓ | MWD ↓ |
|---|---|---|---|---|---|
| NLSB | 0.04 | 0.10 | 0.74 | 0.24 | 0.48 |
| MIOFLOW | 0.09 | 0.28 | 0.79 | 0.388 | 0.66 |
| **DMSB(ours)** | **0.03** | **0.06** | **0.67** | **0.22** | **0.41** |

Table 6: Numerical result of Wasserstein-1 ($W_1$), MMD, energy distance and Max-sliced Wasserstein distance (MWD) on the **velocity** of 5 dimensions single-cell RNA-seq dataset using 500 generative samples and 500 ground truth data.

| Dim=5 | Energy ↓ | MMD ↓ | $W_1$ ↓ | SWD↓ | MWD ↓ |
|---|---|---|---|---|---|
| NLSB[1] | 0.44 | 1.37 | 1.75 | 0.83 | **1.40** |
| MIOFLOW | 0.68 | 2.11 | 1.88 | 0.94 | 1.54 |
| **DMSB(ours)** | **0.40** | **0.85** | **1.67** | **0.74** | 1.43 |

---

[1]See special clarification (Appendix.C) for the velocity generated NLSB

Table 7: Numerical result of Wasserstein-1 ($W_1$), MMD, energy distance and Max-sliced Wasserstein distance (MWD) on the **velocity** of 100 dimensions single-cell RNA-seq dataset using 500 generative samples and 500 ground truth data.

| Dim=100 | Energy ↓ | MMD ↓ | SWD↓ | MWD ↓ |
|---|---|---|---|---|
| NLSB[2] | 2.12 | 1.6 | 0.94 | 1.27 |
| MIOFLOW | 9.18 | 2.41 | 1.89 | 5.66 |
| **DMSB(ours)** | **0.36** | **0.18** | **0.39** | **0.78** |

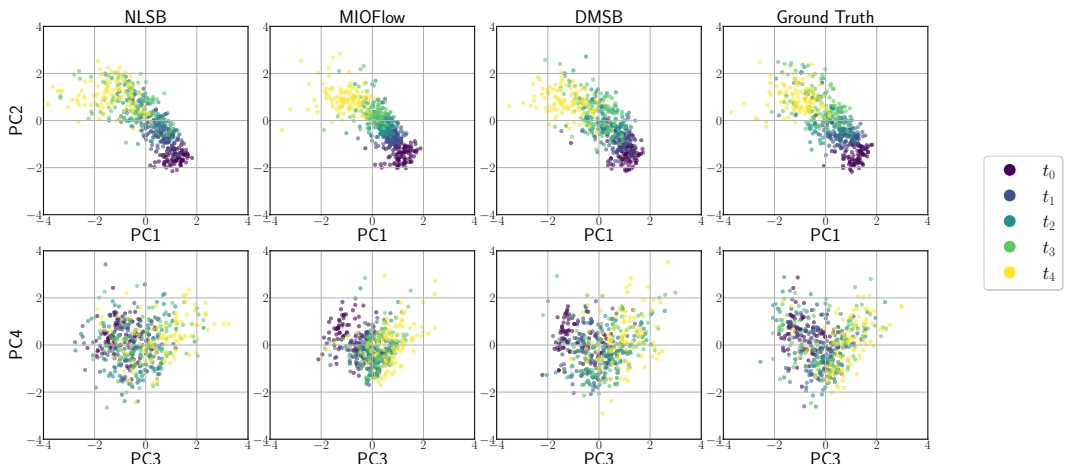

Figure 9: Comparison of population-level dynamics on 5-dimensional PCA space at the moment of observation for scRNA-seq data using MIOFlow, NLSB, and DMSB. We display the plot of the first 4 principle components (PC). All method performs well under this experiment setup.

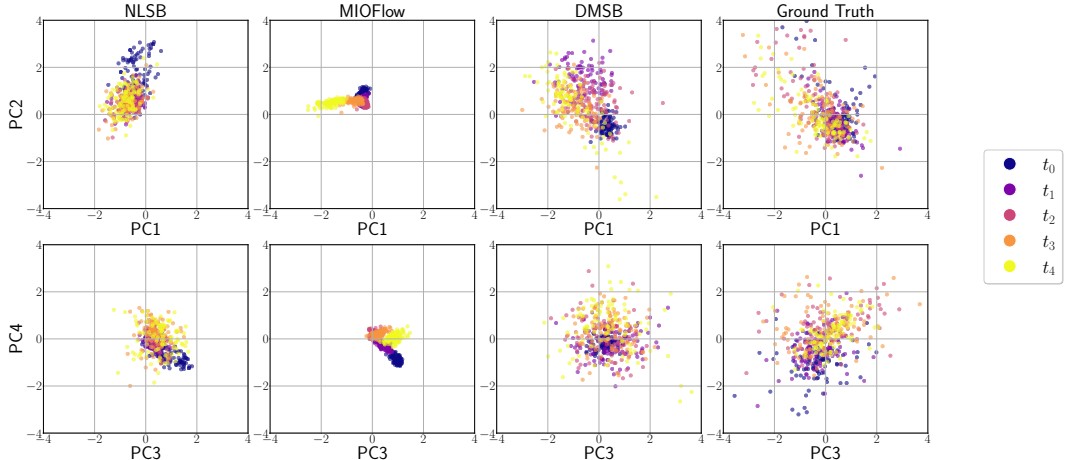

Figure 10: Comparison of estimated velocity on 5-dimensional PCA space at the moment of observation for scRNA-seq data using MIOFlow, NLSB, and DMSB. We display the plot of the first 4 principle components (PC). For the results of NLSB, see special clarification of NLSB in Appendix.C

---

[2]See special clarification (Appendix.C) for the velocity generated NLSB

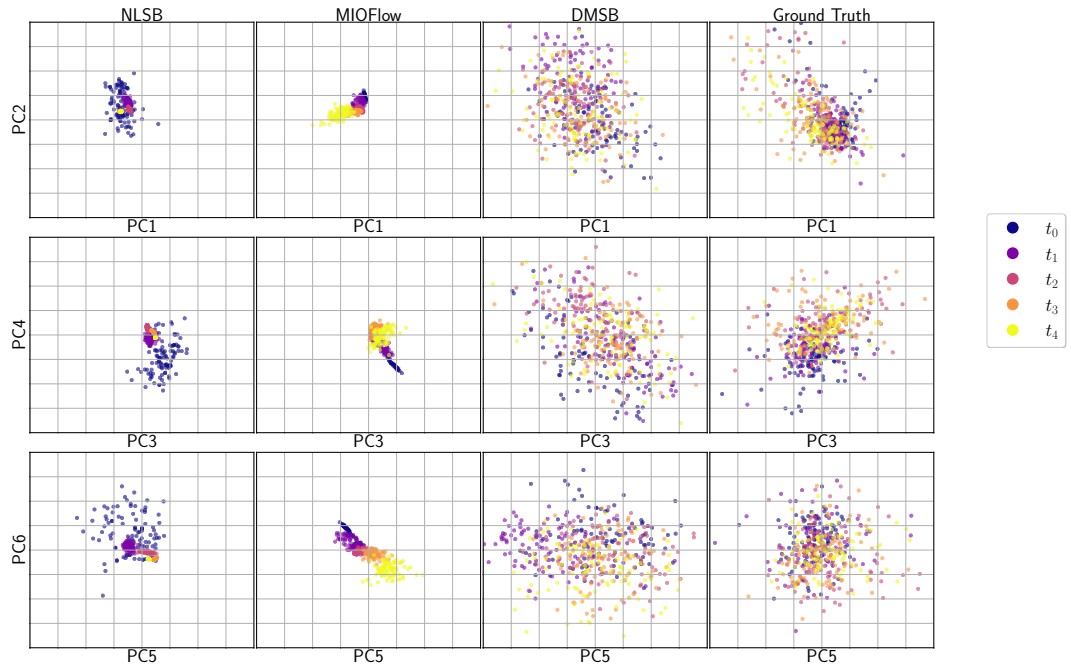

Figure 11: Comparison of estimated velocity on 100-dimensional PCA space at the moment of observation for scRNA-seq data using MIOFlow, NLSB, and DMSB. We display the plot of the first 6 principle components (PC). For the results of NLSB, see special clarification of NLSB in Appendix.C

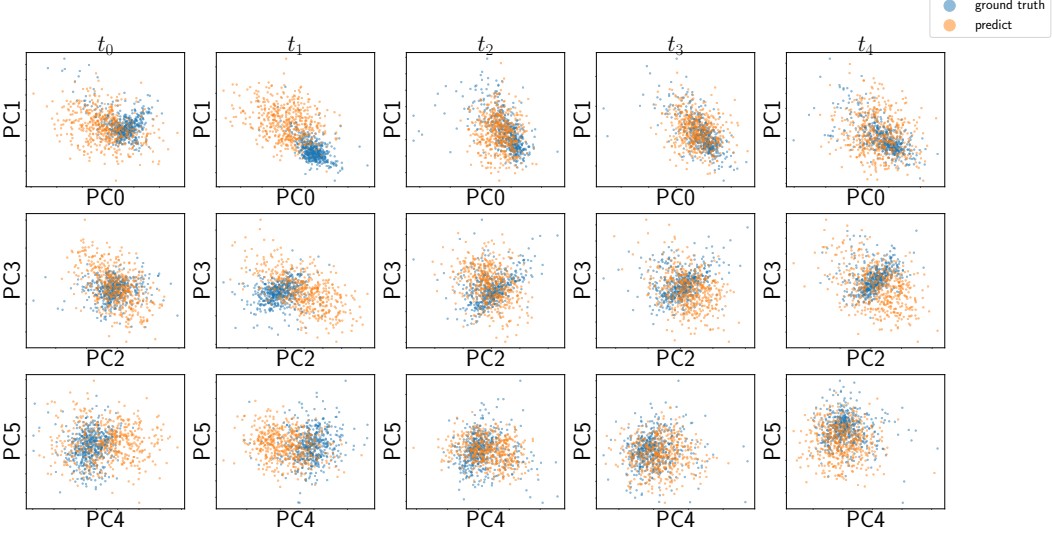

Figure 12: Comparison of estimated velocity on 100-dimensional PCA space at the moment of observation for scRNA-seq data using DMSB with ground truth. We display the plot of the first 6 principle components (PC).

## G   Intuitions of Propositions and Theorems

1. Remark for Proposition 3.1: Within each half-bridge IPF, the variable $\mathbf{Z}_t$(or $\widehat{\mathbf{Z}}_t$) is essentially learning the reverse-time stochastic process induced by $\widehat{\mathbf{Z}}_t$. This process can also be viewed as minimizing the approximated parameterized negative log-likelihood.

2. Remark for Proposition 4.2: In order to matching the reference path measure in KL divergence sense, one need to match both the intermediate path measure eq.(9a) and the boundary condition eq.(9b). In the traditional two boundary SB case, matching the boundary condition is often disregarded due to either having a predefined data distribution or a tractable prior. However, in our specific case, as the velocity is not predefined, it becomes imperative to address this issue and optimize it through the application of Langevin dynamics.

3. Remark for Proposition 4.3: Following the same argument as the remark of Proposition 4.2, it becomes evident that, in this particular scenario, there is no need to account for the data distribution since there are no position constraints when optimizing with $K$. Consequently, the optimal solution will inherently align faithfully with the reverse diffusion and adapt to the boundary conditions imposed by the reference path measure.

4. Remark for Proposition 4.5: We indeed underexplained an important nontrivial fact (thank you so much for catching it): the unique structure of SB leads to a beautiful fact that the score is proportional to the sum of the forward and backward drift terms. This facilitates the sampling of velocity. Specifically, the score function can be obtained using eq(11), as supported by the findings in eq(24). It can also be understood as the one realization of Nelson duality (see Lemma 1 in [46] and [28]).

## H Complexity

Here we provide the complexity of our algorithm.

Table 8: Time complexity w.r.t Dimensionality ( Marginals=5)

| # dimensions | 5 | 10 | 50 | 100 |
|---|---|---|---|---|
| Train | 24min | 25min | 33min | 44min |
| Sampling | 1sec | 1.6sec | 2.0 sec | 2.02sec |

Table 9: Time complexity w.r.t number of marginals (Dim=100)

| # Marginal | 2 | 3 | 4 | 5 |
|---|---|---|---|---|
| Train | 32min | 25min | 33min | 44min |
| Sampling | 2.02sec | 1.6sec | 2.0 sec | 2.02sec |

