# OpenReview forum: "Deep Momentum Multi-Marginal Schrödinger Bridge"
_NeurIPS.cc/2023/Conference — NeurIPS 2023 poster_

### Official Review · Reviewer_EhNn · 2023-07-05

**Soundness:** 3 good
**Presentation:** 3 good
**Contribution:** 3 good
**Rating:** 6
**Confidence:** 4

**Summary:**

The paper tackles the multi-marginal Schrodinger bridge space in phase space, by proposing a computationally tractable solver (DMSB). It leverages alternating Bregman projections to adapt the iterative proportional fitting algorithm in order to deal with multiple convex constraint sets.

The DMSB framework constrains the stochasticity of particle dynamics to their velocity component and therefore ensures smooth trajectories. The authors show how the proposed algorithm better describes scRNA-seq data by reconstructing lower-variance trends in the dynamics and (approximately) recovers the velocity field of particles.


**Strengths:**

The paper builds on well-known results involving Schrödinger bridges (SBs) and convex optimization to practically solve the multi-marginal SB problem while ensuring smooth trajectories. It extends SBs [13] by considering random velocities, rather than random infinitesimal displacements. Furthermore, it generalizes the popular critically-damped Langevin diffusion (CLD) [12] framework to non-gaussian velocity marginals at the extremes.

A key contribution consists in performing 2 alternate optimization stages over the two sets $k_{\text{boundary}}$ and $k_{\text{bridge}}$, containing appropriately-partitioned constraints. The extension of the mean-matching objective to the phase space is also novel but relatively straightforward.

The content is well-motivated and appropriately described. Both mathematical results and algorithmic solutions are accompanied by remarks and informal insight, which help to understand their purpose and significance.
Several tables and diagrams further clarify the presentation by (i) precisely stating the notation used (Table 2), (ii) the problem description (Figure 1), (iii) the comparison against previous work (table 1), and the structure of the algorithm (Figure 2).


**Weaknesses:**

The rationale behind the decoupling of constraints, i.e. the requirement that marginalizing joint densities over the previous/next timesteps yield the same distribution, could be made more explicit to the reader. In particular, it would be advisable to explain why the solution proposed by the authors avoids the “geometric averaging issue” (line 156) which affects instead the solution in [20].

I think that Proposition 4.5 should be followed by a more extensive discussion on the differences between constraints in Eqs. 9b and 10b: Why a conditional distribution appears in the former and a joint one in the latter?

**Miscellaneous**

In addition, I point out minor inaccuracies found while reviewing this draft:
- The quantities optimized in the formulas in the first row of Figure 1 are incorrect ($u$ and $a$ do not appear anywhere in the optimization objective).
- Proposition 4.1 looks more like a definition (and it is not even stated as a proposition in the work cited as a source).
- Wrong sections are cited on the right of Figure 2.
- There are spelling mistakes in lines 34, 66, 89, and 246.
- I would suggest refining the references to the Appendix, by clearly specifying if the link points to a proposition or a section (e.g., in line 128).


**Questions:**

**Experiments**

It is unclear why the colors used in Figure 3 vary across the rows. Could it be possible to identify the position and velocity of each group of points by using the same colors in both graphs?

Contrary to what was stated in the text (“it is almost serendipity that DMSB can also learn the reasonable velocity trajectory without any access to ground truth”, line 230), I don’t find the approximate recovery of velocities surprising. Given the formulation of momentum Schrödinger bridges (lines 125-126), the algorithm must find a reasonable velocity field, in order to produce meaningful trajectories.


**Limitations:**

As stated by the authors, the benefits brought by DMSB are not detectable when learning simple trajectories in relatively low-dimensional spaces. It is therefore unclear whether many practical problems would benefit from it since this comes at the expense of doing without convergence guarantees.

---

> ### Author Rebuttal · Authors · 2023-08-08
>
> # To Reviewer EhNn
> We express our gratitude to the reviewer for their valuable feedback. The summary provided is accurate, and the questions raised are both intriguing and perceptive.
>
> Kindly find below our itemized responses, organized in order to address each of the reviewer's concerns.
> #### **1. In particular, it would be advisable to explain why the solution proposed by the authors avoids the “geometric averaging issue” (line 156) which affects instead the solution in [20].**
> In Section 4 of reference [20] (page 7), the authors introduce an analytic solution that incorporates the geometric averaging of two densities in both the numerator and denominator. The result is based on the constrain set $K_i$ (using notation from [20]), which accounts for the boundary from two directions, leading to the inclusion of geometric averaging in the analytic solution. While it is feasible to estimate this value in the discretized space, addressing this matter in the continuous space poses a challenge, and currently, no optimal solution exists for it.
>
> One of the key highlights of our paper is the successful avoidance of this issue. We achieve this by decomposing the $K_i$ constraints into separate $K_{boundary}$ and $K_{bridge}$ components, allowing us to address and resolve the problem effectively through our proposed approach.
>
> #### **2.  Why a conditional distribution appears in the former and a joint one in the latter?**
> One can obtain similar results as in the proof of Proposition B.5, but the outcome differs from that presented in line 480. Due to the constraints imposed on $K_{boundary}$, it becomes necessary to decompose the joint distribution as $p_{t_i}(x_{t_i})$ and $\hat{q}(v_{t_i}|x_{t_i})$. However, in the optimization of $K_{bridge}$, there are no remaining constraints, indicating that the global minimization of the optimization will precisely match the reverse diffusion while simultaneously meeting the boundary condition.
>
> Interestingly, even though we lack density information from the boundary condition in the reference path measure, we fortuitously possess samples from it, which can be leveraged for constructing the reference path measure for the subsequent iteration. To achieve this, it is essential to ensure that the joint distribution at the boundary is equal (i.e., eq. (42b)).
>
> #### **3 Miscellaneous**
> Thanks for the careful reading! We will update in the revision.
> #### **4 It is unclear why the colors used in Figure 3 vary across the rows. Could it be possible to identify the position and velocity of each group of points by using the same colors in both graphs?**
> We intend to utilize distinct colors to represent position and velocity in the revised version. The draft plot with the desired color scheme can be found in Figure 1 of the attached Rebuttal PDF file.
>
> #### **5 convergence guarantees?**
> Regarding theory, as discussed in the conclusion section, despite the satisfactory performance of our approach, we acknowledge that we currently lack theoretical convergence results. However, we find encouragement in the work of [1], which sheds light on the convergence proof even when the exact solution at each Bregman Iteration is not available. As [1] was almost concurrent, we hope we could consider analysis based on it as a direction for future research.
>
> Regarding empirical performance, we mostly agree with reviewer; however, even in very low dimension, our algorithm still poses marginal advantage as  evidenced by Table 5.
>
> [1]  Provably Convergent Schrödinger Bridge with Applications to Probabilistic Time Series Imputation. arXiv:2305.07247

---

### Official Review · Reviewer_3Whf · 2023-07-05

**Soundness:** 4 excellent
**Presentation:** 3 good
**Contribution:** 3 good
**Rating:** 6
**Confidence:** 3

**Summary:**

The paper aims at solving efficiently in high dimensions the multi marginal momentum Schrödinger Bridge, that is Schrödinger Bridge in phase space with multiple marginal constraints. They also tackle the issue of marginals constraints where only the positions are enforced. They reach this objective by proposing a new algorithm: DMSB.

The paper first focuses on adapting the iterative proportional fitting (IPF) algorithm to the momentum Schrödinger Bridge with fully specified marginals in phase space. They show that the IPF can be seen as an alternating maximization of log-likelihood.

Next they focus on the multi-marginal momentum Schrödinger Bridge. They propose a new formulation of the marginals constraints in order to apply their version of IPF as log-likelihood maximization via policy specification. Moreover they prove that the optimality conditions for the optimal bridge (under their new set of constraints) allows to sample the velocities when they aren't initially specified.

In order to train efficiently in high dimension they adapt a neural network parametrization of the policies known in the classical SB setting to the momentum SB setting.

Finally they test the the performance of DMSB on synthetic data. The performance on high dimensional data is higher than the baselines algorithm. Moreover they are able to recover velocity distributions which is a novelty.

The contributions can be summarized as follows:
- Extend the IPF to the momentum SB
- Introduce a new set of constraints for the Bregman iterations which allows for a neural net parametrization of the policies and sampling of velocities when they are not specified.
- Proposal and benchmark of a new algorithm DMSB made to solve momentum multi-marginal SB






**Strengths:**

The presented use case of their algorithm is cell profiling over time. In that regard the proposed algorithm performs well above baselines and adresses the following technical challenges: sampling of the velocities, curse of dimensionality, partial informations on the marginals.

The sampling of velocities and curse of dimensionality are both tackled by a combination of proposition 3.1 and proposition 4.2/4.3. Proposition 3.1 proposes a novel and interesting formulation of the half bridge using log-likelihood. This is combined with a novel way to deal with the marginal constraints which is a key advance towards efficient high dimensional computing with respect to prior works.

Globally the article puts swiftly together multiple well known ideas and adds key ingredient (decoupling of marginal constraints, log-likelihood IPF) in order to produce an algorithm which removes limitations from prior algorithms. Those limitations being :
- lack of scalability
- need for fully specified velocities
- robustness to missing marginals

Finally the benchmark contains multiple metrics which is helpful in the high dimensional context since the curse of dimensionality renders some metrics less meaningful as they point out.

The explicit description of the algorithm in the appendix is appreciated because it makes the article whole: it contains the theoretical and practical aspects.

**Weaknesses:**

Though inspired by a prior work the section on the neural nets parametrization lacks clarity. How does the log-likelihood minimization is tied to the mean matching objective and thus the loss Lmm? This link is succintly pointed out in the appendix B.5 equation (38).

The training scheme section goes rapidly over how the neural networks are trained. The figure 8 in appendix clarifies a lot the training. The explanation on the discretization method is unclear in section 4.5.

Finally the main advantage of the algorithm DSMB is the scalability however there are no complexity analysis of the algorithm with respect to dimensions and number of marginals.

**Questions:**

How long does training take for scRNA-seq using DMSB compared to MIOFlow and NLSB ?

**Limitations:**

The author address the limitations of the works which are mainly the proofs of the theoretical convergence of the algorithm in the neural nets parametrization case of the momentum multi marginal SB. Proofs which are key to back up the interesting practical results.

They also sharply point out the limitation which is tied to their use case which is the impossibility for mmmSB to model for the death and birth of cells.

---

> ### Author Rebuttal · Authors · 2023-08-08
>
> # To Reviewer 3Whf
> We deeply thank the reviewer for all the comments. The summary is accurate and the questions are interesting and insightful.
>
> Please kindly see our itemized replies below in order to address the reviewer's concern.
> #### **1. How does the log-likelihood minimization is tied to the mean matching objective and thus the loss Lmm?**
>
> Our understanding is that the minimization of the mean matching objective ($L_{MM}$) corresponds to optimizing the lower bound of the log-likelihood, as noted by reviewer 17Zf. This was proved in Proposition B.4.
>
> #### **2. The training scheme section goes rapidly over how the neural networks are trained. The figure 8 in appendix clarifies a lot the training. The explanation on the discretization method is unclear in section 4.5.**
> Owing to constraints on page count, Figure 8 has been placed in the appendix; however, it will be reintegrated into the main body of the paper should additional pages become available.
>
> Regarding discretization, the simplest Euler-Maruyama method has been consistently employed throughout the entirety of this manuscript, although our approach can use better discretization as well.
>
> Specific details pertaining to total time and timesteps can be referenced in Appendix Figure 7. We will further explain it in the main paper.
>
> #### **3 Complexity (Dimension, marginal, time)**
> Thank you for bringing up this matter! We concur that it is important to provide a clear disclosure of the algorithm's complexity within the paper so here we provide the empirical complexity results. These analyses will be integrated into the paper in the future revision.
>
> It is noteworthy that our model demands more time and memory in comparison to previous approaches. However, it is worth highlighting that the superior results we achieve significantly surpass the baseline performance. Here are detailed numerical values. We record the wall time when all of evaluation criterion (MWD,SWD and MMD) does not drop empirically (empirically converges). In terms of sampling, we record the wall time for sampling one forward trajectory with 256 batch size.
>
> All of the results are based on single-cell RNA dataset.
> ##### **Complexity w.r.t Dimensionality (marginals = 5)**
> Dimension | train | Sampling
> --- | --- | ---
> |5     |24 mins|1.62 sec
> |10     |25mins|1.63 sec
> |50     |33 mins|1.84 sec
> |100     |44 mins|2.01 sec
> ##### **Complexity w.r.t Marginals (dim=100), Remark: we keep the same time steps, that leads to almost same sampling time.**
> Number of Marginal | train | Sampling
> --- | --- | ---
> |2     |32mins| 2.01 sec
> |3     |33 mins|2.01 sec
> |4     |38 mins|2.01 sec
> |5     |44 mins|2.01 sec
>
> ##### **Time Complexity Comparison(dim=100, marginals=5)**
> algorithm | DMSB | NLSB  | MIOFlow |
> --- | --- | --- | ---
> |Training     |44 mins|30 mins|20mins
> |Sampling     |2.01 sec|1.6 sec|2.12 sec|

---

### Official Review · Reviewer_A2vV · 2023-07-06

**Soundness:** 4 excellent
**Presentation:** 1 poor
**Contribution:** 3 good
**Rating:** 6
**Confidence:** 4

**Summary:**

In this paper, the authors present an algorithm, DMSB (Deep Multi-Marginal Momentum Schrödinger Bridge), to approximate solutions to an extension of the Schrödinger Bridge (SB) problem into phase space (mmmSB), where (i) marginal constraints on the position are given across time and (ii) stochasticity is only introduced on the velocity variable (which makes trajectories smoother). This framework aims at solving multi-marginal trajectory inference problems, i.e., inferring likely stochastic dynamics of particles on a time interval, given snapshots of them at certain time steps. This work proposes several contributions: (i) it extends the approach proposed by [1] (which solves a single-variable formulation of SB) to phase space (Proposition 3.1), (ii) it presents an efficient numerical scheme for the Iterative Proportional Fitting (IPF)-type algorithm presented by [2] and restated in Proposition 4.1, to solve mmmSB in practice, (iii) it shows great performance on realistic high-dimensional single-cell RNA sequencing ($d=100$) compared to previous baselines [3,4].

[1] Likelihood training of schrödinger bridge using forward-backward sdes theory, Chen et al., 2021.

[2] Multi-marginal Schrödinger bridges, Chen et al., 2019

[3] Manifold interpolating optimal-transport flows for trajectory inference, Huguet et al., 2022.

[4] Neural Lagrangian Schr$\backslash$" odinger bridge, Koshizuka et al., 2022.


**Strengths:**

This work provides a well-motivated and theoretically well grounded adaptation of the SB problem to the multi-marginal setting with smooth trajectories. The authors derive a solid framework and present convincing numerical experiments on real-world data.

**Weaknesses:**

- In my opinion, this paper is not easy to follow and some statements are provided without context or comment, which consists in the main limitation for me at this stage. Since I find the content pretty good, I will increase my score if the authors accept to be clearer in the main paper, by answering my questions and following my general comments given below.

- This paper lacks a theoretical result of convergence (at least non-quantitative, in the ideal setting where the neural networks would perfectly fit the drift terms of the forward and backward SDEs). It is not clear to me how the convergence result stated in [5] extends to the current setting.

[5] Iterative Bregman projections for regularized transportation problems, Benamou et al., 2015.

**Questions:**

- What is the reference measure $\xi$ used in the momentum-like setting (see Line 125 and Eq (6)) ? As far as I understand, it is the path given by the Brownian motion propagated along the velocity variable with the corresponding ODE $d x_t= v_t dt$. The authors should state it explicitly.
- In my opinion, $\bar{\pi}$ is the solution of the problem given in Proposition 4.4 without any other constraint. Does the solution given in the main paper actually include the constraint on the boundary ? What is $q_{t_0}$ in Line 187 ?
- I recommend the authors to specify the set of measures on which the optimization is performed in Propositions 4.1, 4.2 and 4.3: is it a collection of couplings or joint distributions over all the N states ?
- As far as I understand, the constraints given by $K_{bridge}$ enable to smooth out the trajectories of the particles along the whole time interval. To prove the efficiency of their method, could the authors compare their method with the procedure where the only constraints are given by $K_{boundary}$ (which should work too if I am not wrong) ?
- What is the definition of $\mathcal{L}_{reg}$ (Algorithm 4) ?
- How many Bregman iterations did you perform in your experiments ?

Major comments:
- I recommend the authors to give the proof of Proposition B.6 in the appendix, since it is not straightforward.
- For sake of clarity, the authors should introduce in the main paper a consistent notation to refer to couplings (eg, $\pi_{t_1, t_2}$) and joint distributions (eg, $\pi_{t_1:t_2}$) in order to avoid any confusion.

Minor comments:
- The authors should insist more on the (stochastic optimal control) formulation of the SB problem that they are solving in this paper. I don't think it is well highlighted in Figure 1.
- There is inconsistency of the control variable ($a$ or $u$) in Appendix B.1.
- I think there is a typo between $\hat{m}$ and $\bar{m}$ in Algorithms 3 and 4.
- For sake of clarity, I think that the system of PDEs in Eq (20) should be displayed in the main paper.

**Limitations:**

Unlike [1] or [5] where experiments on Celeba dataset ($d\geq3072$) are performed, the dimension does not exceed 100 in the experiments of this paper. In my opinion, it is crucial to present experiments with such order of dimensionality to study the scalability of DMSB.

[1] Likelihood training of schrödinger bridge using forward-backward sdes theory, Chen et al., 2021.

[5] Score-based generative modeling with critically-damped langevin diffusion, Dockhorn et al., 2021.

---

> ### Author Rebuttal · Authors · 2023-08-08
>
> # To Reviewer A2vV
> We extend our sincere gratitude to the reviewer for the valuable comments. Please kindly find below our itemized responses, presented in an effort to address each of the reviewer's concerns.
>
> #### **1. This paper lacks a theoretical result**
> As discussed in the conclusion section, it is important to note that, despite achieving satisfactory performance, we currently lack theoretical convergence results for our approach. However, the recent work of [1] has provided valuable insights into the convergence proof even in scenarios where the exact solution at each Bregman Iteration is unavailable. As that was almost concurrent, we intend to explore and investigate this direction as part of our future research efforts.
>
> [1]  Provably Convergent Schrödinger Bridge with Applications to Probabilistic Time Series Imputation. arXiv:2305.07247
>
>
> #### **2. What is the reference measure ?**
> Thank you for bringing this to our attention. Indeed, your observation is accurate. The path corresponds to the trajectory obtained by propagating the Brownian motion along the velocity variable:
>
> $dx_t=v_tdt$
>
> $dv_t=g_tdW_t$
>
> and it yields analytically available statistics which is aligned with the one proposed in [2] (see eq(12)). The results can be obtained by using the calculation in Section 6.1 of [3] after setting $g_t=\sqrt{\epsilon}$. We will add this and the detailed derivation in the future revision.
>
> [2] Chen et al. "Multi-marginal Schrödinger bridges"
>
> [3] Särkkä et al. "Applied Stochastic Differential Equations"
> #### **3. Is the solution of the problem given in Proposition 4.4 without any other constraint. Does the solution given in the main paper actually include the constraint on the boundary ?**
> Yes, the solution of the problem in Prop 4.4 is without any boundary constraint. Specifically,
> - The solution of Proposition 4.2 in the main paper **does** include the constraint. The constraint is implied in the $\rho_{t_i}$ in the numerator.
> - The solution of Proposition 4.4 in the main paper **does not** include the constraint. Hence, the boundry condition is implied in the $q_{t_0}$ in the numerator which is induced by the reference path measure eq(10b).
>
>
> #### **4. I recommend the authors to specify the set of measures on which the optimization is performed in Propositions 4.1, 4.2 and 4.3?**
> Apologies for any confusion. As presented in the paper, we have decomposed the optimization of Proposition 4.1 into two distinct propositions, namely Proposition 4.2 and Proposition 4.3. In the case of Proposition 4.2, the optimization process involves joint distributions $\pi_{t_i:t_{t+1}}$, considering two marginals at $t_i$ and $t_{i+1}$. Meanwhile, Proposition 4.3 entails the optimization of the joint distribution $\pi_{t_0:t_N}$.
>
> #### **5.Could the authors compare their method with the procedure where the only constraints are given by $K_{boundary}$?**
> We have conducted the experiments per suggestions provided in the review, and we can verify that the suggested algorithm to compare is functional, as evidenced by Figure 2 in the rebuttal PDF. However, importantly its performance is not on par with our proposed approach, particularly in terms of convergence speed and final performance.
>
> #### **6. What is the definition of  $L_{reg}$ (Algorithm 4) ?**
> Sorry for the missing reference. $L_{reg}$ means the regularization mentioned in Ln 210.Specifically, it is the regularization term proposed in [4], which can potentially force the the condition of Prop.4.5 (see line 210).
>
> [4] Ki-Ung Song, "Applying Regularized Schrödinger-Bridge-Based Stochastic Process in Generative Modeling"
>
> #### **7. How many Bregman iterations did you perform in your experiments ?**
> We are using 15 Bregman Iterations for all the experiments and 30 Bregman Iterations for Petal experiment.
>
> #### **8. Provide the proof of Proposition B.6?**
> As we are unable to update the revision at this stage, we provide here a sketch of the proof and will provide a full version in a future revision:
>
> The results can be derived similarly to the proof of Proposition B.5, but with variations beginning from line 480. The constraint in $K_{boundary}$ necessitates decomposing the joint distribution into $p_{t_i}(x_{t_i})$ and $\hat{q}(v_{t_i}|x_{t_i})$. However, in the optimization of $K_{bridge}$, no constraints remain, implying that the global minimization of the optimization will precisely match the reverse diffusion while satisfying the boundary condition. Interestingly, despite lacking density information from the boundary condition of the reference path measure, we fortunately possess samples from it, which can be utilized to construct the reference path measure for the next iteration. To achieve this, it is imperative to ensure that the joint distribution at the boundary remains equivalent (i.e., as given by eq.(42b)).
>
> #### **9 consistent notation to refer to couplings (eg, ) and joint distributions?**
> we apologize for any confusion caused. As per your suggestion, we have rectified the notation in all sections of the rebuttal reply. The updated version will be included in the forthcoming revision of the document.
>
> #### **10. There is inconsistency presented variable and PDE should be put in the main paper**
> Thanks for the careful reading! It is indeed a typo and it is easy to fix. We will update the confusing typo and variable and display the PDE in the main paper once we can update the revision and if we do not exceed the page limit.
>
> #### **11. Image experiments?**
> Regrettably, due to time constraints and limited hardware resources (one Nvidia 3090Ti GPU), we are unable to conduct image experiments during the rebuttal phase. However, we will thoroughly consider this aspect for future investigations should the opportunity arise. Thank you for your understanding and consideration of our constraints.

---

> > ### Comment · Reviewer_A2vV · 2023-08-12
> >
> > Thank you for your response.
> >
> > In particular, the answers to questions 1, 2, 4, 6, 7, 8, 9 and 10 are satisfying. I would like to insist that some technical details on the experiments such as the **time discretization** along the paths (mentioned in the rebuttal to Reviewer 17Zf), the **number of Bregman iterations** and the **regularization term** $L_{reg}$ should appear *clearly* in the revised version of the paper. Overall, the authors also agree that the presentation of the paper may be improved (setting, intuition for the results and their proofs, notation, ...) so that it is easier to read and to understand the content, which is a good point.
> >
> > **About question 5**: I thank the authors for the experiment made on this aspect. I really think that it is of great interest (i) to prove that asymptotically, the constraint $K_{bridge}$ is not needed (if I understand correctly) but also (ii) to show the importance of this constraint in practice, since it smooths out the trajectories. I recommend the authors to include this discussion in the revised version of their paper, to bring even more intuition on their method.
> >
> > However:
> > - **about question 11**: I understand the statement made by the authors, but I still think that the lack of experiments in *real* high-dimensional setting is the main limitation of the paper on the experimental side in its current version. I really think that it would make the contribution stronger if such experiments were conducted !
> > - **about question 4**: unfortunately, **I am still unconvinced** by the explanation given by the authors for the result of Proposition 4.4 and give details about it below.
> >
> > Since the reference measure $\bar{\pi}$ is a path measure along the whole path (i.e., from $t_N$ to $t_0$), we have by definition that the path measures $\bar{\pi}\_{i : i+1}$ and  $\bar{\pi}\_{i-1:i}$ have the same marginal in $t_i$. Then, $\bar{\pi}$ satisfies the constraint $K_{bridge}$ and therefore, the solution to the optimization problem in Proposition 4.4. should simply be $\bar{\pi}$...
> >
> > Given the answer that you gave me (and given the answers given to other reviewers about Proposition 4.4), what I understand from your explanation (**and from Figure 8**) is the following: what you compute at this stage is $\mu^\star=\mu\_{t_0} (\bar{\pi}\_{|t_0})^R$, where $R$ is the time-reversal operator. As far as I understand, $\mu^\star$ is therefore the solution to the KL optimization problem with constraints $K_{bridge}$ and $K_{t_0}$. Hence, I think that the constraint $K_{t_0}$ is missing in Proposition 4.4.
> >
> > - Could the authors explain to me where I am wrong ?
> > - Could the authors explain to me what $q_{t_0}$ refers to ? I did not find any definition of it in the paper...
> >
> > Thank you !

---

> > > ### Author Response · Authors · 2023-08-13
> > > **Response to Reviewer A2vV**
> > >
> > > ##  To Review A2vV:
> > > Thank you for your attentive reading. These questions are right to the point and extremely helpful to us. Glad to hear that a good amount of previous concerns are resolved, and we are delighted to be able to discuss further.
> > >
> > > 1. **Details on discretization, number of Bregman iterations, regularization term should appear clearly in the revised version**: Thank you. We completely agree and will make sure that will be the case.
> > > 2. **I really think that it would make the contribution stronger if such experiments were conducted !**: We totally agree and this is something of high priority in our todo list for further work. Again we apologize for our insufficiency of computational resources.
> > > Just to help us better design such experiments in the future, may we ask what could be a good motivation for considering multi-marginal SB in the pixel space, i.e. each marginal being a distribution of images? Any insight from the reviewer will be deeply appreciated and helpful for us to design the recommended experiments. Thank you!
> > > 4. **The solution to the optimization problem in Prop.4.4 should simply be $\bar{\pi}$:** The reviewer touched upon a subtle but very interesting complication (thank you!) We actually encountered it during the development of our algorithm and thought carefully about it. Interestingly, when formulating the reference path measure, the path measure inherently satisfies $K_{bridge}$ in an "automatic" manner. Nevertheless, as illustrated in Figure 2 of the supplementary PDF, the optimization outcomes fall short of the anticipated level of satisfaction. This discrepancy might be attributed to the absence of a smooth procedure, as astutely pointed out by the reviewer in the "about question 5" section. Recognizing this discrepancy, we set out to devise an additional optimization strategy for $K_{bridge}$ which is not only still theoratically aligned with Bregman Iteration but also heuristically smoothing out the trajectories as the reviewer mentioned. Consequently, the algorithm was refined to its current version. One can observe the improvement from Figure 2 in supplementary PDF. Per your suggestion in the response, we will certainly incorporate this discussion into the revised version.
> > > 5. **What you compute at the stage is $\mu^{*}=\mu\_{t0}(\\bar{\\pi})^R$, and therefore the solution to the KL optimization problem with constrants $K_{bridge}$ and $K\_{t\_0}$:** If we understood your notation correctly, we think you are referring to $\pi^{\star}=\\mu\_{t\_0}(\bar{\pi}\_{|t\_0})^R$ instead of $\\mu^{\star}=\\mu\_{t\_0}(\\bar{\\pi}\_{|t_0})^R$? If so, you are mostly correct. However, what we compute at this stage is $\pi^{\star}=p\_{t\_0}(\bar{\pi}\_{|t\_{0}})^R$ in your notation. $p\_{t\_0}\equiv \hat{q}\_{t\_0}$ where $\hat{q}\_{t\_0}=\int \\bar{\\pi}\_{t\_0,t\_N}d \mu\_{t\_N}$, which will not include the constraint $K_{t_0}$ (which corresponds to $\mu_{t_0}$ in your notation I think?), and it only includes the constraints $K_{bridge}$. Conceptually, in order to find the minimizer of $KL(\pi|\bar{\pi})$ within the constraints $K_{bridge}$, one will not need to include information of $K_{t_0}\in K_{boundary}$.
> > >
> > > 7. **What is $q_{t_0}$?** This should be $\hat{q_{t_0}}$ as being corrected in the reply above. Thank you very much for catching it and we are very sorry for the typo.
> > >
> > > Any further questions are welcome. Your comments greatly helped us to improve the clarity of our work and are deeply appreciated!

---

> > > > ### Comment · Reviewer_A2vV · 2023-08-14
> > > >
> > > > Thank you for your comment !
> > > >
> > > > **About the experiments on image datasets**: I have to admit that I am not particularly aware of such application (although, for example, one may want to infer the evolution of cancer cells based upon snapshots of scanners for health applications, but I acknowledge that I have not background about this). For me, such experiments on image dataset would be firstly *qualitative* (since a ground truth would not be easily available), and one could verify that we recover well the constrained marginals. I think that it would surely demonstrate the scalability of your method when used with complex neural networks such as UNET, and may be first step to real-world applications in generative modeling community.
> > > >
> > > > **About Proposition 4.4**: thank you for the first answer to my question, it helps a lot to understand, and indeed, a discussion should be made about this. Indeed, I was referring to $\pi^\star$ and not $\mu^\star$, sorry for that. About the second part of your answer : as far as I understood, the marginal of the reference measure in Prop. 4.4. at time $t_N$ is actually $\mu_{t_N}$ (for me, it is clear from the first sentence of Proposition 4.4.  but also from Figure 8, see the last line on the left). Therefore, we would have $\int \bar{\pi}\_{t_0, t_N} d \mu\_{t_N}=\bar{\pi}\_{t_0}$. Hence, I still end up with the fact that $\bar{\pi}$ is the solution to the optimization problem of Prop. 4.4 if you don't include the boundary condition $K_{t_0}$ (which actually appears in Fig 8, last line on the left!).
> > > >
> > > > To be clear, I commit myself to increase my score to 6 or 7 if the authors convince me on this last point. Thank you !

---

> > > > > ### Author Response · Authors · 2023-08-15
> > > > > **Response to Reviewer A2vV**
> > > > >
> > > > > ##  To Review A2vV:
> > > > > **Image Experiments** We express our gratitude for your insightful recommendation. As previously stated, it will be high priority in our future work.
> > > > >
> > > > > **About Proposition 4.4**
> > > > > Indeed, the reviewer's assessment is entirely accurate, and the distinction lies solely in the manner of representing the identical  $\pi^{\star}$. Allow us to encapsulate this in the following summary:
> > > > >
> > > > > For the sake of convenience, we will adopt the notation provided by the reviewer. Our aim is to tackle the following optimization problem (assuming, without loss of generality, that $\bar{\pi}$ propagates in the backward time direction: $\mu_{t_N}\bar{\pi}_{|t_N}$):
> > > > >
> > > > > $$\min_{\pi \in K_{bridge}} KL(\pi|\bar{\pi})$$
> > > > >
> > > > > (1) The reviewer suggests that the optimizer of this problem can be implicitly represented by $\pi^{\star}=\\bar{\\pi}=\\mu\_{t\_N}\\bar{\\pi}\_{|t_N}$. We totally agree, and one can notice that this choice is actually functional (Fig.2 in Supplementary PDF).
> > > > >
> > > > > (2) However, this optimizer can also be represented as $\\pi^{\star}=\\hat{\\mu}\_{t\_0}(\\bar{\\pi}\_{|t_N})^{R}=\int \\bar{\\pi}d\\mu\_{t\_N}\\cdot(\\bar{\\pi}\_{|t\_N})^{R}$, which is basically the time reversal of the reference path measure in which samples of $\mu_{t_0}$ can be empirically obtained for free (we are a simulation-based approach).
> > > > >
> > > > > These two formulations yield the same optimizer; however,
> > > > > 1. We have opted to approximate $\pi^{\star}$ in the manner described by formulation (2), which differs from the suggestion put forth by the reviewer.
> > > > > 2. We chose this 2nd representation of the optimizer because it led to *better empirical outcomes*, as previously compared. More precisely, we employed a neural network to approximate the forward drift term, denoted by $z_{t}^{{\theta}}$. In conjunction with the boundary condition $\hat{\mu}_{t_0}$, it allows an approximation of the aforementioned solution. This approximation creates a difference between the practical outcomes of formulations (1) and (2).
> > > > > 3. Most interesting is why this approximation actually led to better empirical results. As black-box neural network training and approximation are involved, we do not have a full theory at this moment. However, our observation and intuitive guess are, our approximation has an implicit bias of smoothing effect, which is beneficial. More precisely, formulation (1) actually admits an analytical solution, but getting it will not be the end of the story because obtaining $\pi^*$ is just one step of the Bregman iterations. The analytical solution is rigid in the sense that connections at the boundaries are not very smooth, and in our experiments, this made the next iteration more difficult. This is because the next iteration will need to use this iteration's result as the reference path measure, and the rigidity makes the training less stable in our experience. Surprisingly, formulation (2) did not exhibit this issue. In summary, our conjecture for the better empirical results of our formulation lies in the smoothing property associated with the optimization process we used to approximate $\pi^{\star}$, via formulation (2), as duly recognized by the reviewer.
> > > > >
> > > > > We hope that this explanation could address the concern raised by the reviewer!

---

> > > > > > ### Comment · Reviewer_A2vV · 2023-08-15
> > > > > >
> > > > > > Thank you for the response, there was a lot to say indeed ! The explanations that you gave me made finally sense to me. As far as I understand, $\bar{\pi}$ is indeed the solution as I said but instead of using it as given, you compute its reversal **starting from its terminal state** (while I thought from the beginning that it was a **constrained marginal**), but correct me if I am wrong!
> > > > > >
> > > > > >  I think that I did not understand this at first sight partially because of the notation : the difference between $\mu_0$ and $\hat{\mu}_0$, was not clear and not well explained in the paper... I really recommend the authors to be particularly careful about it in the revised version of the paper, and about the general notation.
> > > > > >
> > > > > > Moreover, this explanation of the optimization problem is really interesting and not intuitive at all, I think that it should be displayed in the paper. In general, I think that the authors should add some intuition and interpretation to their results, as other reviewers said.
> > > > > >
> > > > > > All in all, the reviewers answered all my questions and promised to improve the reading of their paper. I really think that this paper is a good contribution and therefore I increase my score to 6.

---

> > > > > > > ### Author Response · Authors · 2023-08-15
> > > > > > > **Response to reviewer A2vV**
> > > > > > >
> > > > > > > Thank you for your response!
> > > > > > >
> > > > > > > Your current interpretation aligns accurately with our methodology. Our approach is indeed computing the reversal of $\bar{\pi}$ starting from its terminal state.
> > > > > > >
> > > > > > > We extend our heartfelt apologies for any possible confusion that may have arisen due to the usage of the current notation. In the upcoming revised version of the paper, we are committed to exercising increased diligence with regard to both the specific employed notation and the general notation used throughout the document.
> > > > > > >
> > > > > > > Acknowledging the feedback provided by all the reviewers, we intend to augment our results section with intuitive explanations and comprehensive interpretations.
> > > > > > >
> > > > > > > Once again, we express our gratitude for the valuable and substantial discussion!

---

### Official Review · Reviewer_5VcR · 2023-07-06

**Soundness:** 3 good
**Presentation:** 3 good
**Contribution:** 3 good
**Rating:** 6
**Confidence:** 2

**Summary:**

The paper addresses the topic of multi-marginal trajectory inference in high dimensions using Schrödinger Bridge (SB). In particularly, the authors focus on the so-called momentum SB in phase space where the resulting trajectories in position space are smooth interpolations between the intermediate marginals. The motivation for this is that in real physical systems smooth trajectories are often more likely than abrupted changes in drift direction at intermediate marginals. After an introduction of the preliminaries of Schrödinger Bridge and Bregman iterations, the phase space formulation of SB is introduced, which this work then builds upon. By restructuring the problem formulations and introducing an efficient training scheme for the involved function approximators, the authors show how to solve the multi-marginal momentum SB problem in a computationally efficient way. This is then utilized in the experiments which comprise both a synthetic as well as a 100-dimensional real-world use-case.

**Strengths:**

- Originality: The authors make several important technical contributions that allow for a computationally efficient solution of the (multi-marginal) momentum SB problem in high-dimensional phase spaces. Related work and the foundations the authors build upon are discussed and cited in the manuscript.
- Quality: All original propositions are supported with detailed proofs in the appendix (which I did not check). The experimental results show a convincing improvement over the state of the art. In the conclusion, limitations of the work are addressed.
- Clarity: The paper has a good introduction of the work it builds on. In addition, the novel contributions are clearly stated. The submission is well written and quite accessible on pages 1-4 where the relevant prior work is explained.
- Significance: The presented contributions are likely to enable machine learning practitioners to apply the momentum (multi-marginal) SB method to many interesting high-dimensional trajectory-inference problems. The advances over the state of the art are demonstrated convincingly on both an artificial and a real-world use-case. Further, the approach is also applicable if the marginal velocity distributions are not available, which is probably relevant in many real world use-cases where velocity information can be much harder to obtain the positional information.

**Weaknesses:**

- Clarity: The novel contributions are likely difficult to understand for a non-expert reader. While proofs are presented in great detail in the appendix, a more intuitive explanation or interpretation of propositions 3.1, 4.2, 4.4, 4.5 is lacking. In addition, reproduction and/or extension of the results is probably a significant effort because the source-code is not provided.

**Questions:**

- My main suggestion to the authors would be to make the ideas behind their original propositions accessible to a wider audience by providing some intuition and explanation beyond the mathematical proofs.
- In addition, open-sourcing the code would reduce the entry barrier for practitioners for applying the method.
- In the abstract it reads "In this article, we extend SB into phase space". Is that claim really justified? I had the impression that SB in phase space was introduced in reference 20 and that the novel contribution is in the derivation of a computationally efficient way for half-bridge iterative proportional fitting via negative log-likelihood minimization.

**Limitations:**

Limitations of the presented approach are discussed in the "conclusion and limitations" section.

---

> ### Author Rebuttal · Authors · 2023-08-08
>
> # To Reviewer 5VcR
> We deeply thank the reviewer for all the comments. The summary is accurate and the questions are interesting and helpful.
>
> Please kindly see our itemized replies below for addressing the reviewer's concerns.
>
> #### **1. a more intuitive explanation or interpretation of propositions 3.1, 4.2, 4.4, 4.5 is lacking.**
> We express our gratitude for the thoughtful suggestion, and we intend to incorporate intuitive explanations as well as additional remarks in the future revision. These elucidations will serve to enhance the clarity and comprehensibility of our work, contributing to its overall accessibility. Namely, we are going to add clarification as following:
>
> 1. Remark for Proposition 3.1: Within each half-bridge IPF, the variable $Z_t$ (or $\hat{Z}_t$) is essentially learning the reverse-time stochastic process induced by $\hat{Z}_t$. This process can also be viewed as minimizing the approximated parameterized negative log-likelihood, as correctly pointed out by reviewer 17Zf.
> 3. Remark for Proposition 4.2: In order to matching the reference path measure in KL divergence sense, one need to match both the intermediate path measure eq.(9a) and the boundary condition eq.(9b). In the traditional two boundary SB case, matching the boundary condition is often disregarded due to either having a predefined data distribution or a tractable prior. However, in our specific case, as the velocity is not predefined, it becomes imperative to address this issue and optimize it through the application of Langevin dynamics.
> 3. Remark for Proposition 4.3: Following the same argument as the remark of Proposition 4.2, it becomes evident that, in this particular scenario, there is no need to account for the data distribution since there are no position constraints when optimizing with $K_{bridge}$. Consequently, the optimal solution will inherently align faithfully with the reverse diffusion and adapt to the boundary conditions imposed by the reference path measure.
> 4. Remark for Proposition 4.5: We indeed underexplained an important nontrivial fact (thank you so much for catching it): the unique structure of SB leads to a beautiful fact that the score is propotional to the sum of the forward and backward drift terms (kindly see Lemma B.3 and Line 448 for details). This facilitates the sampling of velocity. Specifically, the score function can be obtained using eq(11), as supported by the findings in eq(24). It can also be understood as the one realization of Nelson duality (see Lemma 1 in [1] and [2]).
>
> [1] Vargas et al. "Solving Schrodinger Bridges via Maximum Likelihood"
>
> [2] Nelson, Edward. "Dynamical theories of Brownian motion."
>
>
> #### **2. In addition, reproduction and/or extension of the results is probably a significant effort because the source-code is not provided.**
> Thanks for the suggestion. We have provided the code-base to the ACs according to the NeurIPS 2023 Rebuttal Instruction. Furthermore, when the paper could be accepted, we will make the code publicly available to facilitate reproducibility and promote transparency in our research. Thank you for your understanding, and we are committed to making the necessary improvements to enhance the quality of our work.
>
> #### **3. In the abstract it reads "In this article, we extend SB into phase space". ,Is that claim really justified? I had the impression that SB in phase space was introduced in reference 20 and that the novel contribution is in the derivation of a computationally efficient way for half-bridge iterative proportional fitting via negative log-likelihood minimization.**
>
> The reviewer is correct and our use of word was imprecise. We will replace it in the abstract by
> "We extend the approach in reference [20] to operate in continuous space". In the main text we will continue this clarification with: "This circumvents the need for expensive space discretization which does not scale well to high dimensions. We also address the challenge of intricate geometric averaging in continuous space setup by strategically partitioning and reorganizing the constraint sets. Furthermore, we enhance the algorithm's computational efficiency by incorporating the method of half-bridge IPF."
> in order to state the contributions of our work more precisely.
>
> The helpful comment is very much appreciated!

---

### Official Review · Reviewer_17Zf · 2023-07-07

**Soundness:** 2 fair
**Presentation:** 2 fair
**Contribution:** 3 good
**Rating:** 6
**Confidence:** 4

**Summary:**

The authors extend the diffusion Schrodinger bridge methodology to mult-marginal setting whereby each marginal is ordered sequentially, in addition, introducing  momentum into the diffusion Schrodinger bridge framework. The method shows excellent performance on trajectory inference tasks.

**Strengths:**

- The proposed work fills a gap in the literature, and the authors demonstrate how diffusion model based Schrodinger bridges [1] and implementation of [2] can feasibly be plugged into the multi-marginal setting [3].
- The method shows excellent performance on trajectory inference tasks, outperforming other methods. The approach feels quite natural.

[1] Bortoli et al https://arxiv.org/abs/2106.01357 \
[2] Chen et al https://arxiv.org/abs/2110.11291 \
[3] Chen, Multi-marginal Schrodinger bridges, 2019

**Weaknesses:**

- The computational complexity of the algorithm is not discussed. It appears multiple (2N reverse diffusions?) must be computed per IPF step. This seems quite costly. I would appreciate discussion and more transparency on this.

- It is very difficult to clearly see what is the main algorithm and training procedure from the main text. I see algorithms are provided in the appendices but these are also not very well detailed in my opinion. The paper would improve in clarity and reproducibility if this was given more attention. This is especially true as no code is provided. Indeed, some training details like caching trajectories are mentioned in passing (as was introduced in [1], which should be cited) but it is not clear from this paper what the authors mean and without a reference it is difficult to know.

- Possible error in likelihood proof / explanations

In line 454, $\mathbb{E}[y_0]$ (and for $\tilde{y}_0$) is decomposed into $\mathbb{E}[y_T]$ minus the path integral of the SDE for $y$. This seems fine. However, my understanding is that unless the diffusion has fully converged to the prior (i.e. after convergence of the IPF, or with infinite regularization in SB), the term  $\mathbb{E}[y_T]=\mathbb{E}[\log p(m_T)]$ is from the terminal point in the diffusion (which is not fixed until after convergence) hence must depend on the network parameters used to define the diffusion. This is commented on in Theorem 2 of [3]. This term is dropped in [2] and in this paper, which makes the likelihood interpretation incorrect for training each step of the IPF.

Instead further work is required, as in [3], to show that time-reversal is a bound of the likelihood and gives **approximate** likelihood training.

The training procedure will still work as it coincides with the time-reversal of a diffusion and the same training procedure as given in [1] but the likelihood explanation does not seem justified as it is currently written.

It appears other reviewers had similar concerns with [2], and also that the overall procedure really is not likelihood training but IPF with time-reversal where each time-reversal can be viewed as approximate likelihood. https://openreview.net/forum?id=nioAdKCEdXB


--------


I am conflicted by this paper. Whilst overall I think it is a good contribution methodologically and believe it can be justified with extra work. I have concerns about clarity and more importantly that this likelihood interpretation is misleading and possibly incorrect.

I would be happy to increase my score if the authors address this. I am happy to be corrected if I misunderstand the likelihood interpretation.

- [1] Bortoli et al https://arxiv.org/abs/2106.01357
- [2] Chen et al https://arxiv.org/abs/2110.11291
- [3] Song et al, Maximum likelihood training of score-based diffusion models, 2021

**Questions:**

- Why use mean matching of [1] rather than divergence based training advocated in [2]? I agree they are theoretically equivalent up to an integration by parts but practically different. Similarly, the authors state the method is based on [2] (e.g. table 1 and line 116) but then if using the training objective given by [1] is it not more accurate to say the method is based on [1]?
- What is the complexity and training time for this method? It appears that one must perform multiple reverse time diffusions per IPF step.

- [1] Bortoli et al https://arxiv.org/abs/2106.01357
- [2] Chen et al https://arxiv.org/abs/2110.11291

**Limitations:**

- The authors discuss lack of theoretical convergence results
- Computational complexity and time to train is not discussed

---

> ### Author Rebuttal · Authors · 2023-08-08
>
> # To Reviewer 17Zf
> We would like to express our sincere gratitude for your valuable feedback and comments. We truly appreciate the time and effort you invested in assessing our submission.
>
> Please kindly see our itemized replies below in order to address the reviewer's concerns.
> #### **1. The computational complexity of the algorithm is not discussed. It appears multiple (2N reverse diffusions?) must be computed per IPF step. This seems quite costly. I would appreciate discussion and more transparency on this.**
> It is a very vital question which we should include in the main paper. The short answer is, yes, it will have 2N reverse diffusions. However, it will only increase marginal computation cost compare with prior simulation-based work [1],[2]. The reason is ---
>
> We discretize time into $S$ steps (where $S$ is set to 200 for the Petal experiment and 400 for other instances). The reference path measure $\\bar{\\pi}\_{t\_i:t\_{i+1}}$ is simulated using $S/(N-1)$ timesteps, given that we have $N$ marginals. To facilitate the training of $z_t^{\theta}$, we must simulate the reference path measure $\\bar{\\pi}\_{t\_i:t\_{i+1}}$ for all $i \\in [0,N-1]$. Consequently, the total number of simulation steps remains at $S$, which is not prohibitively large. However, it is important to note that additional simulations are necessary for the velocity component. As illustrated in Figure 7 of the appendix, this incurs an overhead of (2(two networks inference)x$N$ NFE) when compared to prior work [1],[2].
>
> [1] Valentin et al. "Diffusion Schrödinger Bridge with Applications to Score-Based Generative Modeling"
>
> [2] Chen et al. "Likelihood Training of Schrödinger Bridge using Forward-Backward SDEs Theory"
>
> #### **2. Unclear Algorithm and missing detail of caching trajectorties and source code**
> We genuinely apologize for any confusion caused by the complex presentation of the algorithm. In the revised version, we will provide a clearer and more straightforward explanation of the algorithm to improve its comprehensibility for all readers. Additionally, we will elaborate further on specific techniques, such as the trajectory cache, which are commonly employed in prior works but may not be familiar to a broader audience.
>
> We have provide the codebased to the ACs according to the NeurIPS2023 Rebuttal instruction. Furthermore, when the paper could be accepted, we will make the code publicly available to facilitate reproducibility and promote transparency in our research. Thank you for your understanding, and we are committed to making the necessary improvements to enhance the quality of our work.
>
> #### **3. clarity and more importantly that this likelihood interpretation is misleading and possibly incorrect.**
> Yes, it is misleading and we will add a remark in a revised version to improve the clarity. Our optimization is to find a path measure that best approximates (in KL) the time reversal of the reference path measure obtained in the previous iteration. This optimization approximates likelihood training.
>
> #### **4.Why use mean matching of [1] rather than divergence based training advocated in [2]?**
> This work is inspired by both [1,2] and we think [1] will lead to more scalable training objective since there is no expensive auto-grad computation. We will update the Line 116 to better state the importance of [1].
>
> [1] Valentin et al. "Diffusion Schrödinger Bridge with Applications to Score-Based Generative Modeling"
>
> [2] Chen et al. "Likelihood Training of Schrödinger Bridge using Forward-Backward SDEs Theory"
>
> #### **4.Complexity?**
> Thank you for bringing this matter to our attention! We concur that it is important to provide a clear disclosure of the algorithm's complexity within the paper, so here we provide some real numbers. These reports will be integrated into the paper in the future revision.
>
> It is noteworthy that our model demands more training time in comparison to previous approaches. However, it is worth highlighting that the superior results we achieve significantly surpass the baseline performance. Here are detailed numerical values. We record the wall time when all evaluation criteria (MWD,SWD and MMD) no longer drop (i.e. empirically convergent). In terms of sampling, we record the wall time for sampling one forward trajectory with 256 batch size.
>
> All of the results are based on single-cell RNA dataset.
>
> ##### **Complexity w.r.t Dimensionality (marginals = 5)**
> Dimension | train | Sampling
> --- | --- | ---
> |5     |24 mins|1 sec
> |10     |25mins|1.6 sec
> |50     |33 mins|2.0 sec
> |100     |44 mins|2.02 sec
> ##### **Complexity w.r.t Marginals (dim=100), Remark: we keep the same time steps, that leads to almost same sampling time.**
> Number of Marginal | train | Sampling
> --- | --- | ---
> |2     |32mins| 2.02  sec
> |3     |33 mins|2.02  sec
> |4     |38 mins|2.02  sec
> |5     |44 mins|2.02  sec
>
> ##### **Training Time Comparison(dim=100, marginals=5)**
> algorithm | DMSB | NLSB  | MIOFlow |
> --- | --- | --- | ---
> |Training     |44 mins|30 mins|20mins
> |Sampling     |2.02 sec|1.6 sec|2.12 sec|
>
>
> #### **4. The authors discuss lack of theoretical convergence results**
>
> As stated in the conclusion section, while our performance results are promising, we acknowledge that the theoretical convergence analysis is currently lacking. However, a recent work [3] addressed the convergence proof in scenarios where the exact solution at each Bregman Iteration is not readily available. As that was almost concurrent, we hope we could consider it as a direction for future research.
>
> [3] Provably Convergent Schrödinger Bridge with Applications to Probabilistic Time Series Imputation. arXiv:2305.07247

---

> > ### Comment · Reviewer_17Zf · 2023-08-21
> >
> > Thank you for the response, it has addressed my concerns provided the stated changes are implemented.
> >
> > I will raise my score to weak accept.

---

### Author Rebuttal · Authors · 2023-08-08

# To All Reviewers
We thank all reviewers for their valuable comments. We are excited that the reviews identified the novelty of our contribution, appreciated our experimental validations and acknowledging the significancy of our work.

The common criticisms rather came from insufficient complexity analysis and inacurate description for certrain concepts, about which we apologize.

In a revised version, we will integrate the clarifications suggested by all the reviews to make the paper easier to be understood, together with additional experimental validations suggested by the reviewers.

An itemized summary for each reviewer is listed below, which in our humble opinion resolves all raised concerns. We sincerely thank you for your time and help in all cases.

## Summary of Revision for all Reviewers
- In alignment with NeurIPS 2023's code disclosure guidelines, we have provided the Area Chair with the anonymous GitHub link to our codebase. Additionally, we have included the command line instructions necessary to replicate the results in the paper, along with an estimate of the expected waiting time.
- We will add the complexity analysis as the reviewers suggest.
- We will release the code when the paper is accepted.
- We will correct the inaccurate statement of the Likelihood as being suggested by reviewer 17Zf.
- We will update the notation for the coupling and the joint distribution. Specifically, $\pi_{t_i,t_{j}}$ will represent for the coupling of two marginals, and $\pi_{t_i:t_{j}}$ will be the joint distribution (path measure) as being suggested by reviewer A2vV.
- We will correct all the typos mentioned by reviewers. Thanks again for your careful and helpful reviews.

---

### Decision · Program_Chairs · 2023-09-21

**Decision:**

Accept (poster)

**Comment:**

The paper deals with the problem of multi-marginal Schrödinger Bridge (SB), employing a phase space formulation, bridging some gap in the literature.

While all reviewers concur that the approach introduced in the paper is indeed novel, interesting, and well-supported, my assessment reveals that there is room for enhancing the paper's clarity. Furthermore, the recommended changes provided by the reviewers should be meticulously incorporated into the final version of the manuscript. I strongly stress to the authors the significance of giving due attention to these aspects and ensuring they are not disregarded.